**Subject Category:**
Biology (whole organism)

ecology/genetics

aquaculture, management, genetics, salmon lice, salmonid, migration

**Author for correspondence:**
A. C. Harvey
e-mail: alison.harvey@hi.no

# Inferring Atlantic salmon post-smolt migration patterns using genetic assignment

A. C. Harvey[1], M. Quintela[1], K. A. Glover[1,2], Ø. Karlsen[1], R. Nilsen[3], Ø. Skaala[1], H. Sægrov[4], S. Kålås[4], S. Knutar[1] and V. Wennevik[1]

[1]Institute of Marine Research (IMR), Bergen, Norway
[2]Institute of Biology, University of Bergen, Bergen, Norway
[3]Institute of Marine Research (IMR), Tromsø, Norway
[4]Rådgivende Biologer AS, Bergen, Norway

ACH, 0000-0001-8422-8763

Understanding migratory patterns is important for predicting and mitigating unwanted consequences of environmental change or anthropogenic challenges on vulnerable species. Wild Atlantic salmon undergo challenging migrations between freshwater and marine environments, and the numbers of salmon returning to their natal rivers to reproduce have declined over several decades. Mortality from sea lice linked to fish farms within their seaward migration routes is proposed as a contributing factor to these declines. Here, we used 31 microsatellite markers to establish a genetic baseline for the main rivers in the Hardangerfjord, western Norway. Mixed stock analysis was used to assign Atlantic salmon post-smolts caught in trawls in 2013–2017 back to regional reporting units. Analyses demonstrated that individuals originating from rivers located in the inner region of the fjord arrived at the outer fjord later than individuals from middle and outer fjord rivers. Therefore, as post-smolts originating from inner rivers also have to migrate longer distances to exit the fjord, these data suggest that inner fjord populations are more likely to be at risk of mortality through aquaculture-produced sea lice, and other natural factors such as predation, than middle or outer fjord populations with earlier exit times and shorter journeys. These results will be used to calibrate models estimating mortality from sea lice on wild salmon for the regulation of the Norwegian aquaculture industry.

# 1. Introduction

Migratory species often travel large distances between habitats, usually between breeding and feeding grounds. The success of these journeys may be closely linked to the fitness and abundance of wild populations and is often associated with key environmental conditions or triggers for optimal timing. Therefore, population viability may be negatively affected by environmental changes in the migration route, for example, due to anthropogenic influence in habitats and ecosystems, or climate change [1].

Studies investigating migration dynamics typically involve the use of mechanical or electronic tagging. Mechanical tags rely on recapturing an individual and provide limited information on migration between tagging and recapture locations, while electronic tags may send periodic information wirelessly from the tag to a receiver [2]. Recent studies have also used population genetic tools to investigate migration patterns in several migratory species [3,4], also coupled with tagging [5,6]. While genetic differentiation is low in some species, for example in birds and bats [7,8], genetic methods have been successful in differentiating between distinct populations of several fish species [9–11], and thus display considerable potential for identification of migrating individuals to their populations of origin.

When fish populations are segregated into reproductively isolated stocks, genetic markers can be used to identify the different allelic frequencies among the different stocks [5]. Genetic stock identification (GSI) estimates either the origin of the individual fish within a mixed sample, known as individual assignment, or proportions of different stocks in a mixed sample, known as mixed stock analysis (MSA), by comparing the individual genotypes in the mixed sample to a baseline with known genotypes [6]. GSI is widely used to uncover the underlying contributions of different populations to mixed-stock fisheries [12], and has been used to identify genetically distinct population units in mixed marine fisheries [13,14], mixed salmonid fisheries [15–19], historical fisheries or on oceanic feeding grounds [10,20,21] and even within rivers [22].

Atlantic salmon (*Salmo salar*) is an anadromous fish native to both sides of the North Atlantic and displays a complex life-history that involves transitions between freshwater and marine environments, as well as challenging long-distance migrations. Adults reproduce in freshwater, with juveniles remaining in the river environment for up to eight years, before undergoing an adaptation to saltwater known as smoltification [23]. The so-called smolts then migrate out of the river into the sea to feed and mature [23]. Seaward migration represents a critical life-stage, and is characterized by high mortality [23]. The initial timing of migration varies among river populations, possibly as a response of adaptations to within-river and marine environmental conditions [24,25], and there is a critical window of downstream migration which provides the best match between arrival time in the sea and optimal environmental conditions that maximize survival and recruitment [25,26]. Therefore, the duration and timing of smolt migration can have a significant effect on the marine survival of salmon populations, with several elements of the immediate environment having a cumulative negative influence on survival, such as predation, exposure to pathogens and parasites, varying food availability and negative anthropogenic effects [23,25]. In recent years, the numbers of wild salmon returning to rivers throughout much of its native range have declined [27]. In Norway, genetic interactions from escaped farmed salmon [28] and sea lice linked to fish farms are the two most significant challenges from aquaculture to wild populations [29].

Norway is currently the world's largest producer of farmed salmon, which are reared in sea-cages located in sheltered coastal areas, resulting in a significant increase in the number of hosts available to sea lice, and in particular, the salmon louse (*Lepeophtherius salmonis*) [30]. This has led to an increase in the abundance of sea lice in coastal areas, with a corresponding increase in lice levels observed on wild salmonids in aquaculture-intensive areas [29,31–33]. Recently (2017), the Norwegian government introduced new regulations for future expansion of the salmonid aquaculture industry [34]. Industry growth is dependent on the estimated additional mortality of wild salmonids that is directly attributable to sea lice within the 13 production zones along the Norwegian coastline [35,36]. Each zone is classified as either 'green', 'orange' or 'red' depending on the estimates of sea-lice-induced mortality, where green zones may expand production, production is frozen at the current level in orange zones, and reduced in red zones [34]. The mortality estimates are calculated by combining a hydrodynamic model which estimates larval dispersal, and a smolt migration model which estimates the infection pressure on virtual smolts moving through the fjord out to sea [37]. The models are calibrated using data collected from field surveillance of lice levels on wild salmonids consisting of

monitoring lice levels on fish in sentinel cages [36,38], post-smolts caught in coastal bag nets [35], and migrating post-smolts caught by trawling within the fjords [37].

Although currently untested, genetic methods to identify post-smolts caught in the coastal field surveillance trawls back to their rivers or regions of origin could provide more information on spatio-temporal migration patterns during the early marine phase. Such investigations, however, require fine-scale genetic baselines of the contributing populations. Datasets have been established for Atlantic salmon populations covering rivers in their western Atlantic range [10,39] and areas in the eastern Atlantic [18,20,40–42]. Here, we developed a genetic baseline representing all the significant Atlantic salmon populations in the Hardangerfjord, a farming intense region in western Norway where sea-lice-induced mortality has been reported to be high [43], to identify the river or region of origin of post-smolts caught in the annual sea lice trawl surveys in the period 2013–2017. The dataset from the present study will be used as further calibration in the virtual smolt migration model, which is discussed elsewhere [37]. In the present study, our overall aim was to investigate the potential of identifying post-smolts back to river or region of origin using MSA, and thereafter, to investigate spatial and temporal patterns of post-smolt migration in this fjord.

## 2. Methods

### 2.1. Sample collection and sampling

#### 2.1.1. Sampling fish in rivers (establishing the genetic baseline)

The samples of fish from rivers, which represent the genetic baseline samples, originated from four main sources: (i) scale or tissue samples that were donated by anglers or by other research institutions or environmental consultancies, (ii) existing genetic data from previously genotyped river samples from other studies conducted by the Institute of Marine Research (IMR) [42], (iii) tissue samples collected previously for other analyses by IMR, and (iv) samples collected by electrofishing during the summer of 2017 and the winter of 2018. The baseline consisted of 1364 individuals from 14 rivers covering a period from 2011 to 2018 (table 1 and figure 1). The River Guddalselva, located within Hardangerfjord, was not included in the baseline, as until very recently it did not contain a native population of Atlantic salmon and was only used in other comparative studies which were based on the differences between farmed and wild salmonids [44,45]. Two rivers, Oselva and Tysseelva, which do not belong to Hardangerfjord but are in the vicinity (figure 1), were incorporated to the baseline as outliers for control. It is assumed that since these rivers are not located within Hardangerfjord, they should not appear in the assigned trawl samples.

#### 2.1.2. Sampling post-smolts in the sea (trawl samples)

Post-smolts were collected in the late spring and early summer from 2013 to 2017 by trawling in the Hardangerfjord for up to four continuous weeks per year (table 1 and figure 1). Captured post-smolts were sedated, examined for sea lice, measured (wet weight, total and fork length), killed using an overdose of Benzocain anaesthetic or a blow to the head and then frozen for subsequent analysis. The number of post-smolts included in the genetic MSA per year ranged from 60 (2013) to 245 (2017). The 2014 trawl samples were not included in further analyses due to low numbers (table 1).

### 2.2. Genotyping

DNA analysis of all samples was conducted at IMR in Bergen during the period 1 May 2017–1 March 2018. DNA was extracted from either fin clips or scales in 96-well plates using the Qiagen DNeasy 96 Blood & Tissue Kit with two negative controls. In total, 31 microsatellite markers were amplified in five PCR multiplexes (microsatellite information (electronic supplementary material, table S1) and amplification protocols (electronic supplementary material, table S2) are found in electronic supplementary material, file 1). PCR products were resolved on an ABI 3730 Genetic Analyser and sized using a 500LIZ size standard (Applied Biosystems). GeneMapper v. 5.0 was used to score alleles manually. A second laboratory technician quality checked each individual sample to ensure scoring accuracy before exporting the data for statistical analysis. Individuals with more than 30% missing alleles were removed from the dataset.

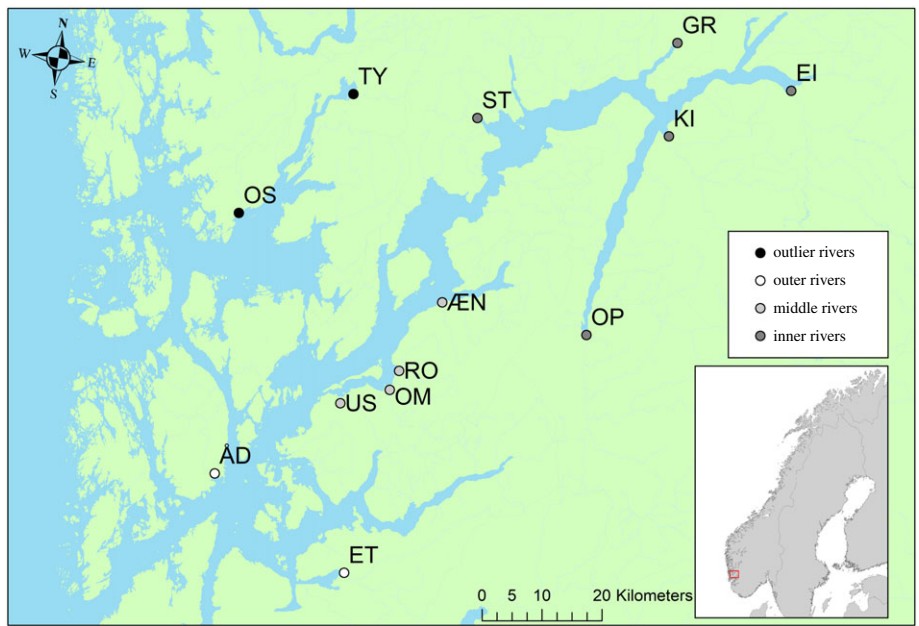

**Figure 1.** Map of Hardangerfjord indicating the rivers included in the genetic baseline. The rivers contained within the regional reporting assignment units of inner, middle, outer and outlier regions are denoted within the legend.

## 2.3. Statistical analysis

### 2.3.1. Baseline genetic structure and assessment

Standard methods were used to characterize the genetic baseline (electronic supplementary material, file 1). In brief, the total number of alleles and allelic richness of each river were calculated, and pairwise $F_{ST}$ was compared among rivers and for temporal samples within rivers that were more than three years apart. STRUCTURE [46] and principal component analysis (PCA) using GenoDive v. 2.0b23 [47] were used to further elucidate population genetic structure. Assignment analyses were carried out at the river level and to regional reporting units (see below).

The assignment accuracy of the baseline was assessed using the Leave One Out test in ONCOR [48]. The test removes one fish sequentially from a baseline population and then estimates their origin using the rest of the baseline. Following on these results, the rivers were grouped into regional reporting units. The regional grouping consisted of four units (inner, middle, outer and outlier fjord populations, table 1 and figure 1) based primarily on the geography of the fjord system. The accuracy of the baseline based on regional reporting units was also assessed using the Leave One Out test as above and by MSA. The MSA was carried out by 100% simulations and realistic fishery simulations using the Anderson method [49] in ONCOR [48]. The 100% simulations involved simulating random samples of each baseline population and assigning them back to the full baseline. The simulations were based on 1000 simulations of 200 fish per baseline population and simulated reference sample sizes that were the same as the dataset. The realistic fishery simulations involved randomly selecting fish from each baseline population and assigning them back to the truncated baseline. The realistic fishery simulations were based on 1000 simulations of 200 fish per baseline population and performed using two mixtures: one with equal proportions of fish from each regional reporting unit and one with proportions based on the estimated smolt production predicted per river that was previously calculated as part of an assessment of the ecological status of rivers within the Hardangerfjord in 2008 (Outlier: 0, Inner: 0.36, Middle: 0.35; Outer: 0.29) [50–52]. Mixture proportions were calculated using a maximum-likelihood method where the genotype frequencies were calculated using the method of Rannala & Mountain [53]. Confidence intervals (CI) were calculated using 1000 bootstraps.

### 2.3.2. Assigning the post-smolts

A MSA in ONCOR was used to estimate the mixed stock composition of the post-smolt trawl catches based on assignment to regional reporting units. Mixture proportions were calculated using conditional maximum likelihood [54], and bootstrapping of 1000 iterations was performed to estimate

**Table 1.** Summary data for the baseline rivers and the post-smolts samples. Source indicates where the samples came from, IMR, Institute of Marine Research; RB, Rådgivende Biologer AS. Stocking indicates whether the river is stocked with eggs or fry from hatcheries. *: trawling did not occur continuously during these dates in 2013 and 2014. Reporting units consist of rivers that are situated in proximity to each other and within certain areas in the fjord system.

| river code | river name | reporting Unit | no. genotyped | no. samples | no. alleles | allelic richness | year sampled | source | stage | stocking | trawl dates |
|---|---|---|---|---|---|---|---|---|---|---|---|
| ÅD | Ådlands-vassdraget | outer | 58 | 56 | 10.07 | 8.70 | 2018 | IMR | juvenile | NA | |
| ET | Etneelva | outer | 149 | 148 | 13.63 | 10.35 | 2013 | IMR | adult | yes | |
| US | Uskedalselva | middle | 222 | 200 | 13.53 | 10.01 | 2016, 2017 | IMR | adult, juvenile | no | |
| OM old | Omvikelva old | middle | 93 | 90 | 12.07 | 9.71 | 2011, 2012 | IMR & RB | juvenile | no | |
| OM new | Omvikelva new | middle | 141 | 120 | 13.00 | 10.13 | 2016–2018 | IMR & RB | juvenile | no | |
| RO | Rosendalselvane | middle | 90 | 80 | 12.63 | 10.26 | 2017 | IMR | juvenile | no | |
| ÆN | Æneselva | middle | 43 | 35 | 10.20 | 9.05 | 2014 | RB | juvenile | no | |
| OP | Opo | inner | 111 | 107 | 12.37 | 9.8 | 2013, 2014 | RB | juvenile | yes | |
| KI | Kinso | inner | 99 | 80 | 11.63 | 9.27 | 2018 | IMR | juvenile | yes | |
| EI | Eidfjord-vassdraget | inner | 118 | 99 | 11.93 | 9.52 | 2014, 2017 | IMR & RB | juvenile | yes | |
| GR | Granvins-vassdraget | inner | 101 | 89 | 12.23 | 9.84 | 2018 | IMR | juvenile | historic | |
| ST | Steinsdalselva | inner | 60 | 60 | 11.53 | 9.8 | 2017 | IMR | juvenile | yes | |
| OS | Oselva | outlier | 98 | 73 | 11.07 | 9.04 | 2015, 2016 | IMR & RB | juvenile | historic | |
| TY | Tysseelva | outlier | 150 | 127 | 10.87 | 8.92 | 2014, 2015 | RB | juvenile | no | |
| | Trawl | | 291 | 245 | 15.03 | 11.91 | 2017 | IMR | smolt | | 8 May–3 Jun |
| | Trawl | | 236 | 180 | 14.63 | 11.83 | 2016 | IMR | smolt | | 1 May–12 Jun |
| | Trawl | | 141 | 125 | 13.97 | 11.68 | 2015 | IMR | smolt | | 30 Apr–10 Jun |
| | Trawl | | 27 | 3 | | | 2014 | IMR | smolt | | 6 May–12 Jun* |
| | Trawl | | 119 | 60 | 11.87 | 10.55 | 2013 | IMR | smolt | | 29 Apr–16 Jun* |

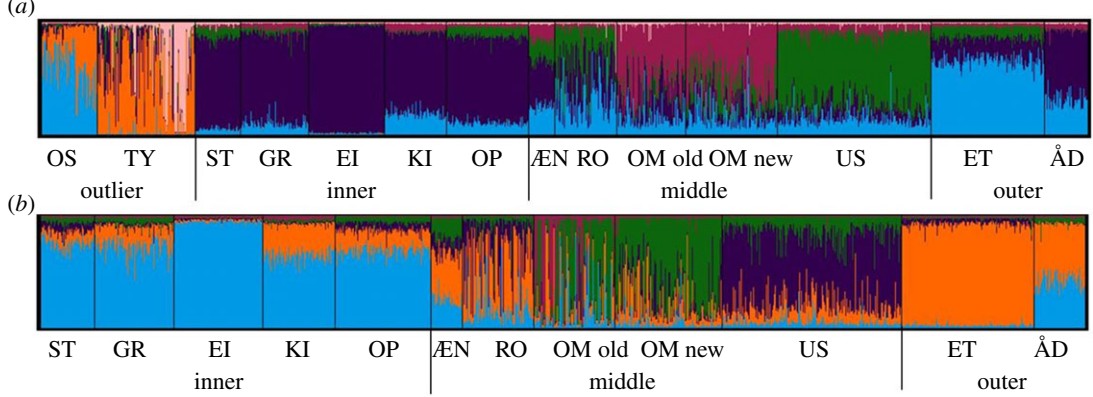

**Figure 2.** Bar plot clusters of the STRUCTURE outputs for the full baseline (14 rivers) $K = 6$ ($a$), and for the baseline with the outlier populations (Oselva and Tysseelva) removed where $K = 5$ ($b$). Populations are grouped into the regional assignment units.

95% CI. The MSA was undertaken for each year and for each week within each year. Weekly MSA results per year were examined visually to investigate temporal patterns of assignment among the regional reporting units.

# 3. Results

## 3.1. Baseline genetic structure and assessment

Significant genetic differentiation was observed between all rivers, as revealed by pairwise $F_{ST}$ (electronic supplementary material, table S3). The River Omvikelva was the only river in the baseline with temporal samples separated by more than 3 years. These displayed very small but significant genetic differences ($F_{ST} = 0.005$, $p < 0.000$).

When examining the STRUCTURE results, for the baseline containing all 14 populations, Evanno's test [55] and StructureSelector [56] indicated that $\Delta K$ was highest when $K = 6$ (figure 2$a$) while the highest $\Delta K$ for the truncated baseline (12 populations—removing the rivers Tysseelva and Oselva, which are geographical outliers) was given as $K = 5$ (figure 2$b$). For both analyses (including either 14 populations or 12 populations after the outlier rivers were removed), the inner rivers clustered together, in agreement with the geographical approach for regional assignment. The rivers designated to the middle and outer regions clustered more distinctly as separate river populations; however, it was decided to base the assignment on geographical regions as described above. The PCA analysis identified genetic structure that to a large degree overlapped with the clustering analysis provided above by STRUCTURE (electronic supplementary material, figure S1).

The Leave One Out test applied to examine accuracy of assignment within the genetic baseline found that on average 53.1% of fish were correctly assigned back to their river of origin (electronic supplementary material, table S4). In most cases, mis-assignment occurred between rivers located near to each other, further supporting the use of regional assignment units over individual rivers. When using regional reporting units (i.e. inner, middle and outer fjord), self-assignment accuracy improved to 72.1% on average (table 2). The MSA found the assignment accuracy for the regional baseline to be robust. In the 100% simulations, the CI ranges of the estimated proportions contained the actual proportions used in the simulations for all regional reporting units apart from the outer region (electronic supplementary material, figure S2). In the realistic MSA simulations, the CI ranges of the estimated proportions contained the real fishery proportions used in the simulations for all regional reporting units (electronic supplementary material, figure S2).

## 3.2. Assigning the post-smolts to regional units

The weekly MSA for each year exhibited a general temporal trend of higher estimated proportions of fish from rivers located in the outer and middle regions of the fjord being present in the trawl catches in the earlier weeks than fish from the inner region (figure 3). In the later weeks of each year, the proportions of fish from rivers located in the inner region of the fjord increased compared to the outer and middle

**Table 2.** Proportion of the baseline samples that were assigned to each regional assignment unit by the Leave One Out test in ONCOR. The diagonal (in italics) represents the proportion of individuals that were correctly self-assigned to each region.

|          | outlier | inner | middle | outer |
| -------- | ------- | ----- | ------ | ----- |
| outlier  | *0.74*  | 0.06  | 0.09   | 0.11  |
| inner    | 0.02    | *0.80* | 0.11  | 0.07  |
| middle   | 0.04    | 0.15  | *0.67* | 0.13  |
| outer    | 0.05    | 0.12  | 0.16   | *0.68* |

regions (figure 3). These trends were evident in all trawl years. Figures from 2013, 2015 and 2016 are presented in electronic supplementary material, figures S3–S5).

# 4. Discussion

To our knowledge, this is the first study to infer fjord migration timing of wild Atlantic salmon post-smolts using genetic methods. Our results showed that, on average, post-smolts originating from rivers located in the outer and middle regions of the fjord migrate out of the fjord before post-smolts originating from rivers located in the inner region of the fjord. We demonstrate that genetic methods can, therefore, be used to infer spatial and temporal patterns of post-smolt migration in the coastal marine environment.

## 4.1. Genetic assignment success

Among the baseline rivers, there was some evidence of hierarchical genetic structuring (figure 2; electronic supplementary material, figure S1). Furthermore, assignment accuracy to geographical region was substantially higher than to individual river of origin as mis-assignments tended to favour geographically close rivers. Assignment to regional reporting units provided a trade-off between accuracy and precision in order to identify the temporal and spatial migratory patterns of the fish.

The Hardangerfjord is Norway's most aquaculture-intensive coastal region, and multiple rivers within the Hardangerfjord have been admixed with escaped farmed salmon [57–60]. In turn, this has caused a reduction in population genetic differentiation with time due to a homogenizing effect [57,58]. Our baseline samples were sourced from recent years to control for any potential long-term changes in population genetic structure among rivers resulting from introgression of farmed salmon or any other stochastic challenges. The use of additional markers and incorporating non-genetic river-specific information in future assignment studies may also improve the assignment accuracy of the baseline.

The rivers Oselva and Tysseelva were genetically distinct to the rivers inside the Hardangerfjord and were estimated to contribute little to the MSA. This result was intuitive, as they are located outside the fjord, and were only included in the baseline to investigate the accuracy of assignments. It was therefore not expected that any of the post-smolts would originate from either Oselva or Tysseelva. However, in 2016, the MSA estimated a very small proportion of fish belonging to the outlier region. In 2016, trawling occasionally occurred between Stord island and the mainland, therefore it is possible that fish from the River Oselva, which is the second-largest population in the region, occasionally use this channel to migrate to sea.

## 4.2. Spatio-temporal patterns of post-smolt migration

The present study illustrates how genetic tools may be used to infer patterns of migration in salmon post-smolts in coastal regions. In turn, this has provided new information on the migratory dynamics of salmon from the different rivers. The majority of previous knowledge pertaining to migration routes stems from telemetry studies (but see [5,6]). Such studies are often limited by low sample sizes and high costs, and it is rarely feasible to tag fish from all contributing rivers simultaneously. Also, the tagging procedure may have an effect on the behaviour of the tagged fish [2,61].

Across all years of the present study, higher proportions of salmon originating from rivers in the inner region of the fjord were found in later trawl catches, strongly suggesting that the majority of these fish arrive at the outer fjord later than salmon originating from rivers in the middle and outer regions

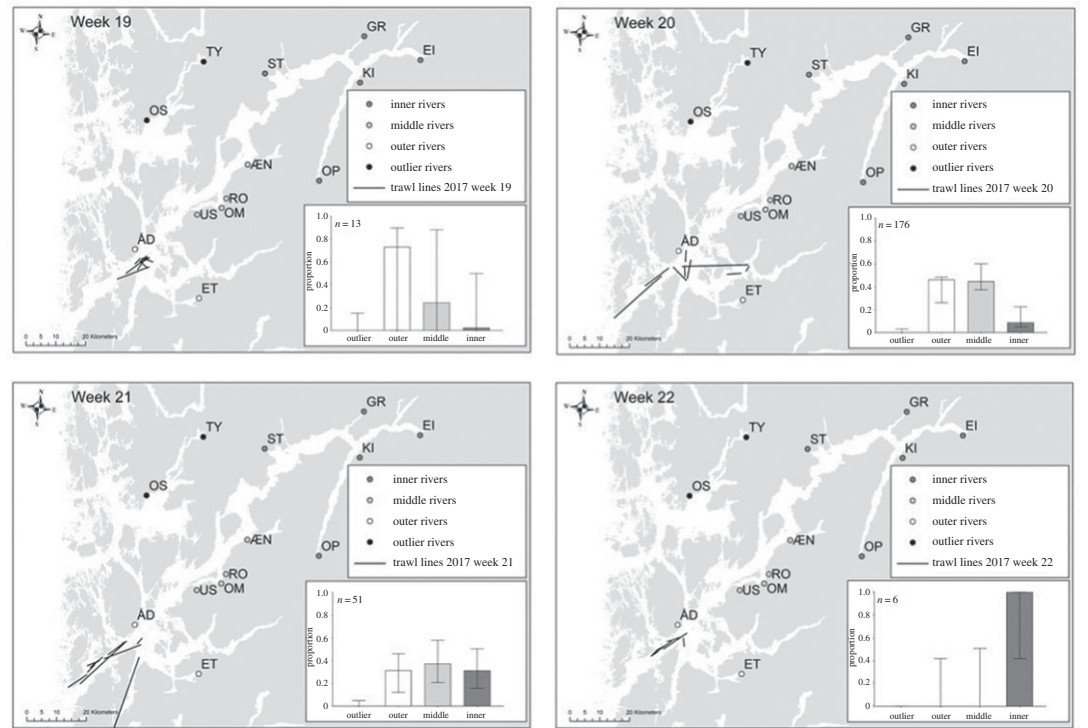

**Figure 3.** Weekly proportions of post-smolts estimated by mixed stock analysis caught in the 2017 trawl survey, including trawl lines for each week. Note that trawls did not cover land, the lines were generated using start and stop coordinates only.

(figure 3). Our results are in agreement with Vollset *et al.* [62], who used mark-recapture data to estimate arrival times to the outer fjord of post-smolts from several rivers on the west coast of Norway. They found that post-smolts from rivers with longer migration distances (inner rivers) would exit the fjord three to four weeks after post-smolts from outer rivers. Differences in fjord exit times may have large population-level implications if fish arrive out of sync with the optimal survival conditions, for example during periods with a high frequency of predators or periods of low food availability [63]. Several studies suggest that populations with longer fjord migrations may be more exposed to mortality-inducing factors within a fjord [62,64,65]. Migration periods that coincide with periods of increased sea lice prevalence, for example through spring and summer [66], may cause high mortality for a particular migration cohort or population. Our data indicate that post-smolts originating from rivers located in the inner region of the fjord face a longer migration and arrive at the outer fjord later than post-smolts originating from rivers located in the middle or outer regions of the fjord. Thus, these populations may experience a higher and longer exposure and infestation pressure from sea lice and other factors affecting marine mortality than the outer populations.

The current regulation of the growth of the aquaculture industry in Norway depends on the estimated mortality of wild post-smolts from sea lice originating from fish farms [36,67]. These estimates are based on models, one of which estimates the spatio-temporal overlap between sea lice and post-smolts in the fjords. Our results highlight the need to incorporate temporal and spatial migration patterns among rivers and regions within a fjord system into these models. The Hardangerfjord is Norway's most aquaculture-intense coastal region, and high infestations of lice on seaward migrating salmon post-smolts has resulted in the additional sea-lice-mediated mortality within this region (production zone 3) being estimated as above 30% (high) in the Norwegian sea lice monitoring programme [43]. Models predicting the overlap between post-smolts and sea lice should take regional differences in infection pressure into account when estimating the additional mortality from sea lice within the Hardangerfjord and other fjord systems with similar geography.

Ethics. Permission for the trawling activities within the Hardangerfjord were granted by The Norwegian Environment Agency (Miljødirektoratet) in accordance with the Act on Salmon Fish and Inland Fish of 15 May 1992 (No. 47 §13, cf. Nature Diversity Act, §18) for each year: 2013 (2013/5291 ART-FF-SJ); 2014 (2014/4225), 2015 (2015/2742); 2016 (2016/3535); and 2017 (2017/33890). Permission for the electrofishing activities within the rivers in Hardangerfjord was granted by Fylkesmannen i Hordaland in accordance with the Act on Salmon Fish and Inland Fish of 15 May

1992 (No. 47 §13, cf. Nature Diversity Act, §18) for 2017 and 2018 (2016/2105 443.1). In addition, the welfare and use of animals was performed in strict accordance with the Norwegian Animal Welfare Act of 19 June 2009, enforced on the 1 January 2010. All personnel involved had undergone training approved by the Norwegian Food Safety Authority.

Data accessibility. The dataset supporting this article has been uploaded as part of the electronic supplementary material.

Authors' contributions. A.C.H. participated in field data collection, carried out the molecular work, participated in data analysis and drafted the manuscript, M.Q. participated in data analysis, K.A.G. conceived and designed the study and participated in data analysis, Ø.K. conceived and designed the study and coordinated the field data collection, R.N. coordinated and participated in field data collection, Ø.S. coordinated field data collection, H.S. participated in data analysis, S.K. participated in field data collection and carried out the molecular work, V.W. conceived and designed the study, participated in data analysis. All authors contributed critically to the drafts and gave final approval for publication.

Competing interests. The authors have no competing interests to declare.

Funding. This project is funded by the Norwegian Ministry of Trade, Industry and Fisheries.

Acknowledgements. We thank everyone who contributed to the project, including those who collected the trawl samples (coordinated by Ø.K.), contributed baseline samples (Rådgivende Biologer AS & the Institute of Marine Research (IMR)), helped to sample the fish and contributed to the genotyping in the laboratory. We also thank Eric Verspoor and John Gilbey for their helpful insights into the data analyses.

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
