## [Reviewer comments · Royal Society Open Science]

Review History

RSOS-190426.R0 (Original submission)

Review form: Reviewer 1

Is the manuscript scientifically sound in its present form?

Yes

Are the interpretations and conclusions justified by the results?

No

Is the language acceptable?

Yes

Is it clear how to access all supporting data?

No

Do you have any ethical concerns with this paper?

No

Have you any concerns about statistical analyses in this paper?

Yes

Recommendation?

Major revision is needed (please make suggestions in comments)

Comments to the Author(s)

Harvey et al. report on post-smolt migration patterns of Atlantic salmon in the Hardangerfjord in Norway. The authors present a genetic baseline for the major Atlantic salmon populations in the Hardangerfjord (based on 31 microsatellites) and attempt to assign post smolt salmon caught in the fjord (4 years worth of data) back to their native rivers/regions. Their data indicate that salmon from rivers draining into the inner fjord regions arrive at the outer reaches of the fjord later, compared to populations from middle and outer fjord river populations. They further investigate the timing of migration of the populations and discuss the implications of their findings with respect to current regulations in salmon production.

I congratulate the authors to a well written manuscript. The size of the dataset (especially for the genetic baseline) is quite impressive, the paper is interesting and I think the analyses are generally sound (but see below). In my opinion, the key resources presented by the paper are the genetic baseline for the major Atlantic salmon populations in the Hardangerfjord, the evaluation of assignment success of post-smolts to their rivers/regions of origin, and the results regarding the arrival times of the populations in the outer fjord region.

I fully agree that more research is needed and every effort should be made to refine regulations for salmon production in the fjord to reduce the effect of industry-mediated parasite pressure on wild salmon smolts. However, the inference regarding migration times as well as the discussion of the implications of these are in my opinion a bit superficial. The authors adopt a migration rate from the literature and calculate the time of departure of salmon based on this and the shortest distance (actually it's not even clear to me which distance they chose) between the river of origin and the location where the fish were caught. I think they should either be done/discussed more thoroughly or removed from the manuscript. I do think this manuscript would be fine without these parts given the impressive amount of data it presents. Halttunen et al. 2018 (ref. 50 in current study) found that individual variability in progression rate and route choice affected by hydrographic conditions had strong effects on fjord residency in Atlantic salmon. Additionally the study found that most fish did not swim directly to the outer straights.

I also recommend that the authors increase their efforts with respect to the reproducibility of their analyses, i.e. sharing of data - see below.

The journal targets a general readership, so mainly people outside the Norwegian salmon production industry. Therefore, I would recommend that the authors give a bit more detail about the current regulations and the models, which were used to inform those. As it is now, the authors report that there are models that predict temporal and spatial overlap between sea lice larvae and post-smolts, and that these are used to 'regulate the production level of the aquaculture industry'. I think for a general readership it would be interesting to know what that means in practice. Do certain production sites in certain parts of the fjord have to be shut down completely over a period, or reduce the number of fish produced, currently when, etc.? Perhaps the authors could even give an example of a production site for which the current regulations dictate that it is shut down in the period from month A to month B and discuss how this strategy is at odds with their findings.

The extent the authors decide to elaborate on this further may of course depend on how much of the focus on the implications for the regulations they decide to retain in a revised manuscript.

Other general comments:

The authors have chosen the colors green, blue, red to represent the outer, middle and inner fjord regions. I suggest to consider readers with red-green color blindness and change this color scheme.

Reproducibility:

- line 155: The analyses are based on 31 microsatellites. The authors provide essentially no information on them except for the IDs in the supplementary table 2. To me it's not clear if they were published previously or are new, whether the authors present the multiplexing assay for the first time, etc. I don't think providing information on conditions on request is sufficient and suggest that the authors provide details (references for published microsatellites, primer sequence in case of new microsatellites, PCR/multiplexing conditions, etc.) in a supplementary table/document, if not in the main text.

- In supplementary File S2, baseline regions tab – please add a column with river name to the table

- In supplementary File S2, trawl all years tab – please add a column with the assignment results for the individuals, e.g. river/region assigned to, probability, unassigned, etc.

- If you decide to retain this part, please provide the minimum distance by sea between the capture point and river of origin you used to calculate the migration times in a supplementary table.

Comments by line, typos, suggestions:

Lines 58-60:

Last sentence of the abstract: 'These results are directly implemented ..' When I first read the abstract I was expecting that the current study presents a recalibrated model, which is not the case. I suggest to remove this sentence or rephrase it towards 'the new data should be incorporated into the existing models'.

114:

we developed an genetic
should be 'developed a genetic'

130:

which anaesthetic?

132:

I suggest to start a new sentence here: 'The 2014 trawl samples were not included ..

148:

"Assignment analyses were carried out.." this sentence should be removed from the section about establishing the genetic baseline towards the next section (around line 169).

155:

In total, 31 microsatellite markers were amplified in five PCR multiplexes (amplification conditions are available on request).

Need to provide references, primers and conditions as supplementary table/document. See comment on reproducibility.

168:

"STRUCTURE and PCA were used to ... and identify potential regional reporting units..". I suggest to remove/change this sentence. It seems to me that the regional reporting units are defined mainly based on geographical location in the fjord. I am not saying that these as defined here are necessarily inappropriate, the self-assignment test based on these certainly shows that – it just doesn't seem to me that it's fair to say the regional reporting units are based on the microsat data. Looking at the structure plot, especially the middle reporting unit is quite a mix

and I wonder if an objective clustering criterion (without a priori geographical information) would group the populations together. Perhaps the PCA shows a clearer pattern, but it is not displayed properly in the supporting file, so I can't assess that.

Lines 176-182:

The 'leave-one-out' test is reasonably straight forward, but I have to admit that it's not entirely clear to me how the MSA works for assessing the accuracy of the baseline. I assume that the general reader won't be either, so I'd suggest that the authors extend this part slightly, also with respect to the subtleties of the different simulations used for the assessment.

202:

to "back-calculate" the approximate.. Perhaps use 'infer'?

224-228:

See comment above. I fail to see that the structure analysis 'largely agrees' with the regional groupings. Perhaps rephrase this and/or discuss in more detail. I think it's fine for the 'inner' group, but especially for the middle and also the outer group I am less convinced. Perhaps the PCA is clearer there, but it's not displayed properly.

Lines 234 - 238:

"For the various MSA fishery simulations, the estimated proportions matched well with the given proportions used in the simulations (Figure S2). In the 100% simulations, the estimated proportions contained the simulated proportions for all regional reporting units apart from the outer region. In the realistic MSA simulation, the estimated proportions matched the simulated proportions well for all regional reporting units."

This sounds all a bit vague to me. 'matched well', 'estimated proportions contained the simulated proportions' - and I am not sure how exactly these findings are informative to judge the accuracy of your baseline. What are the 'estimated proportions' and 'given proportions' here? I assume that one of the two come from your data, but which one? Presumably the 'estimated'? Unless you manage to summarize the results from the assessment of the baseline accuracy from MSA here in a more quantitative manner, I suggest to exclude this part from the main text and move to supplementary material. As is, the result from the 'leave-one-out' test are much more informative.

I am not sure what Figure S2 actually shows - see comment regarding MSA for assessment of the baseline and the simulations in the methods above. There is a typo in part C of the figure - it reads 'simulation proportions'.

Line 241:

'Using individuals assigned to regional units ..' How many percent of individuals per trawl could be assigned to regional units based on your 80% probability criterion? I think this would be worthwhile reporting and I can't seem to find these results anywhere.

Line 248-258:

I am not sure what your point is here and I suggest to work on this part. You are comparing predictions from 2008, which are based on the ecological condition of the rivers, against your results, which are also estimates (and have quite some error bars). Are you meaning to verify your results, by saying that they are close to the predictions from 2008, or are you trying to make a point that the prediction overestimated the contributions of the inner fjord populations, because they did not take into account sea-lice pressure in the fjord caused by the industry or otherwise? In any case, whatever patterns you are trying to highlight here for the inner and middle region, they are reversed in the year 2013, which you fail to mention. So, either keep your summary more general and let the reader decide themselves based on the figure, or discuss the results more thoroughly - why do you think 2013 fell out, for example?

Line 259-261:

"While there was a trend of fish from some of the inner rivers migrating earlier than fish from the middle and outer rivers in some years, overall, there was overlap in the estimated date upon entry to saltwater among the rivers (Figure 5)."

I have already mentioned my concerns regarding these analyses above. Also, I have to say that I don't really see the trend for 'some rivers', either. The only population that comes out

consistently earlier in your analyses is Opo, so if anything I would speak of one population. But, Ref 50 has found lots of individual variation in migration patterns especially for this river..

Line 261: "entry to saltwater" - Please rephrase or define clearly. where does 'saltwater' start. I guess, where the river enters the fjord. In any case, you haven't really calculated this though right? You caught fish somewhere in the fjord and calculated the time salmon spent between their rivers of origin and this place in the fjord.

265-269:

I would remove point (2) from the conclusion as a main finding - see concerns above. Also, for a main result I think it's really very vaguely put: "salmon from different rivers tended to leave their rivers around the same time".

274 - 301.

I think this part of the discussion is nice and could be kept also if you decide against retaining the analyses regarding time of departure from the rivers.

296-297:

"Sea lice prevalence tends to increase over the summer months.." Are there no more specific data available than this? Also, reference needed here. "Over the summer months" - the inner fjord populations tend to arrive a little later - that's true. But it's a matter of +/- one-two weeks. End of May at the latest, that's not yet summer, when sea lice prevalence tends to increase. The thing would have been to check for the extent of sea lice infestation in the trawls, combined with your genetic inference of river of origin..

348

I suggest 'sea-lice mediated mortality'.

350

'clearly indicate that post-smolts originating from rivers located in the inner region of the fjord face a longer migration' - I would disagree there. I don't think it's so clear.

364:

".. genetic approaches do not involve handling or tagging". I disagree there and suggest to remove this statement. In your case (genetic study) of course there was no tagging, but definitely 'handling' of fish. Unless you get your genetic data from eDNA or alike there's always going to be handling.

364-366:

For me, the last sentence is somewhat out of place. Sure, it would be good to combine all available approaches and so on to best understand what's going on. From my point of view you could already discuss telemetry/tagging studies in combination with genetic studies (your work here). Ref 50 even looked at the same rivers, but you don't put your results in context with theirs. I think, if you want to call for a multidisciplinary approach the key thing would be to combine genetic approaches, like yours presented here, with parasitological investigations, to see if fish from inner fjord populations actually show higher infestation rates with sea lice.

Figure 1:

I suggest to remove the trawl lines from the figure - it's impossible to make out specific lines - and adjust the legend accordingly. The individual trawl lines per year are also illustrated in the figures in supplementary file1 - this is enough in my opinion as far as visualization goes. Since you have the data you could also provide the start-, endpoints of the the trawl lines as supplementary data.

Line 547 - identificatino to identification.

Figure 5:

What are the points in figure 5? Are these the individual salmon that you were able to assign back to the rivers?

Supplementary file 1 –

line 22: change 'stricture' to 'structure'

Figure S1 is not displayed correctly.

Review form: Reviewer 2

Is the manuscript scientifically sound in its present form?

No

Are the interpretations and conclusions justified by the results?

Yes

Is the language acceptable?

No

Is it clear how to access all supporting data?

Yes

Do you have any ethical concerns with this paper?

No

Have you any concerns about statistical analyses in this paper?

Yes

Recommendation?

Major revision is needed (please make suggestions in comments)

Comments to the Author(s)

Please see attached file (Appendix A).

Decision letter (RSOS-190426.R0)

07-May-2019

Dear Dr Harvey,

The editors assigned to your paper ("Inferring Atlantic salmon post-smolt migration patterns using genetic assignment") have now received comments from reviewers. We would like you to revise your paper in accordance with the referee and Associate Editor suggestions which can be found below (not including confidential reports to the Editor). Please note this decision does not guarantee eventual acceptance.

Please submit a copy of your revised paper before 30-May-2019. Please note that the revision deadline will expire at 00.00am on this date. If we do not hear from you within this time then it will be assumed that the paper has been withdrawn. In exceptional circumstances, extensions may be possible if agreed with the Editorial Office in advance. We do not allow multiple rounds of revision so we urge you to make every effort to fully address all of the comments at this stage. If deemed necessary by the Editors, your manuscript will be sent back to one or more of the original reviewers for assessment. If the original reviewers are not available, we may invite new reviewers.

- Data accessibility

<http://datadryad.org/submit?journalID=RSOS&manu=RSOS-190426>

- Competing interests

- Authors' contributions

All submissions, other than those with a single author, must include an Authors' Contributions section which individually lists the specific contribution of each author. The list of Authors

should meet all of the following criteria; 1) substantial contributions to conception and design, or acquisition of data, or analysis and interpretation of data; 2) drafting the article or revising it critically for important intellectual content; and 3) final approval of the version to be published.

- Acknowledgements

- Funding statement

on behalf of Dr Kristina Sefc (Associate Editor) and Kevin Padian (Subject Editor)
openscience@royalsociety.org

Associate Editor's comments (Dr Kristina Sefc):

The manuscript has now been seen by two reviewers. Both highlight the value of this impressive dataset, but both are also quite critical about some aspects of the study. In particular, the reviewers emphasize the need to adapt the manuscript to a general (non-fisheries) readership. Please follow their advice, i.e. distinguish between the actual study (genetic assignment of post-smolts) and possible applications, and in connection with the latter, provide the readers with more information about the salmon industry.

Furthermore, please follow the reviewers' suggestions regarding analyses and interpretation of the data. If you decide to retain the analyses of migration times in the manuscript (see reviewer 1), please make sure to discuss potential uncertainties and caveats.

Additionally, please take care to provide the requested information about the microsatellite loci / genotyping.

From my own reading of the manuscript:

Table 1: why is number of samples often lower than number genotyped?

Table 1: Is sample type in any way important with regard to the study? If you retain the column, don't use a sample type category "DNA" – all of the other samples (fins, scales etc.) were used to

extract DNA, and some tissue must have been collected from the Etneelva fish for DNA extraction as well.

In order to determine whether the Structure analysis indeed supports the regional assignments (inner/middle/outer), you should investigate the results for different values of K . With $K < 5$, does the assignment approach a reflection of the three regions?

line 103: sea lice induced mortality

Fig. 5 lacks a color legend for the inner/middle/outer color codes. The legend of Fig. 4 is quite challenging.

Comments to Author:

Reviewers' Comments to Author:

Reviewer: 1

Comments to the Author(s)

Harvey et al. report on post-smolt migration patterns of Atlantic salmon in the Hardangerfjord in Norway. The authors present a genetic baseline for the major Atlantic salmon populations in the Hardangerfjord (based on 31 microsatellites) and attempt to assign post smolt salmon caught in the fjord (4 years worth of data) back to their native rivers/regions. Their data indicate that salmon from rivers draining into the inner fjord regions arrive at the outer reaches of the fjord later, compared to populations from middle and outer fjord river populations. They further investigate the timing of migration of the populations and discuss the implications of their findings with respect to current regulations in salmon production.

I congratulate the authors to a well written manuscript. The size of the dataset (especially for the genetic baseline) is quite impressive, the paper is interesting and I think the analyses are generally sound (but see below). In my opinion, the key resources presented by the paper are the genetic baseline for the major Atlantic salmon populations in the Hardangerfjord, the evaluation of assignment success of post-smolts to their rivers/regions of origin, and the results regarding the arrival times of the populations in the outer fjord region.

I fully agree that more research is needed and every effort should be made to refine regulations for salmon production in the fjord to reduce the effect of industry-mediated parasite pressure on wild salmon smolts. However, the inference regarding migration times as well as the discussion of the implications of these are in my opinion a bit superficial. The authors adopt a migration rate from the literature and calculate the time of departure of salmon based on this and the shortest distance (actually it's not even clear to me which distance they chose) between the river of origin and the location where the fish were caught. I think they should either be done/discussed more thoroughly or removed from the manuscript. I do think this manuscript would be fine without these parts given the impressive amount of data it presents. Halttunen et al. 2018 (ref. 50 in current study) found that individual variability in progression rate and route choice affected by hydrographic conditions had strong effects on fjord residency in Atlantic salmon. Additionally the study found that most fish did not swim directly to the outer straights.

I also recommend that the authors increase their efforts with respect to the reproducibility of their analyses, i.e. sharing of data - see below.

The journal targets a general readership, so mainly people outside the Norwegian salmon production industry. Therefore, I would recommend that the authors give a bit more detail about the current regulations and the models, which were used to inform those. As it is now, the

authors report that there are models that predict temporal and spatial overlap between sea lice larvae and post-smolts, and that these are used to 'regulate the production level of the aquaculture industry'. I think for a general readership it would be interesting to know what that means in practice. Do certain production sites in certain parts of the fjord have to be shut down completely over a period, or reduce the number of fish produced, currently when, etc.? Perhaps the authors could even give an example of a production site for which the current regulations dictate that it is shut down in the period from month A to month B and discuss how this strategy is at odds with their findings.

The extent the authors decide to elaborate on this further may of course depend on how much of the focus on the implications for the regulations they decide to retain in a revised manuscript.

Other general comments:

The authors have chosen the colors green, blue, red to represent the outer, middle and inner fjord regions. I suggest to consider readers with red-green color blindness and change this color scheme.

Reproducibility:

- line 155: The analyses are based on 31 microsatellites. The authors provide essentially no information on them except for the IDs in the supplementary table 2. To me it's not clear if they were published previously or are new, whether the authors present the multiplexing assay for the first time, etc. I don't think providing information on conditions on request is sufficient and suggest that the authors provide details (references for published microsatellites, primer sequence in case of new microsatellites, PCR/multiplexing conditions, etc.) in a supplementary table/document, if not in the main text.
- In supplementary File S2, baseline regions tab – please add a column with river name to the table
- In supplementary File S2, trawl all years tab – please add a column with the assignment results for the individuals, e.g. river/region assigned to, probability, unassigned, etc.
- If you decide to retain this part, please provide the minimum distance by sea between the capture point and river of origin you used to calculate the migration times in a supplementary table.

Comments by line, typos, suggestions:

Lines 58-60:

Last sentence of the abstract: 'These results are directly implemented ..' When I first read the abstract I was expecting that the current study presents a recalibrated model, which is not the case. I suggest to remove this sentence or rephrase it towards 'the new data should be incorporated into the existing models'.

114:

we developed an genetic
should be 'developed a genetic'

130:

which anaesthetic?

132:

I suggest to start a new sentence here: 'The 2014 trawl samples were not included ..

148:

"Assignment analyses were carried out.." this sentence should be removed from the section about establishing the genetic baseline towards the next section (around line 169).

155:

In total, 31 microsatellite markers were amplified in five PCR multiplexes (amplification conditions are available on request).

Need to provide references, primers and conditions as supplementary table/document. See comment on reproducibility.

168:

“STRUCTURE and PCA were used to ... and identify potential regional reporting units.. “. I suggest to remove/change this sentence. It seems to me that the regional reporting units are defined mainly based on geographical location in the fjord. I am not saying that these as defined here are necessarily inappropriate, the self-assignment test based on these certainly shows that – it just doesn’t seem to me that it’s fair to say the regional reporting units are based on the microsat data. Looking at the structure plot, especially the middle reporting unit is quite a mix and I wonder if an objective clustering criterion (without a priori geographical information) would group the populations together. Perhaps the PCA shows a clearer pattern, but it is not displayed properly in the supporting file, so I can’t assess that.

Lines 176-182:

The ‘leave-one-out’ test is reasonably straight forward, but I have to admit that it’s not entirely clear to me how the MSA works for assessing the accuracy of the baseline. I assume that the general reader won’t be either, so I’d suggest that the authors extend this part slightly, also with respect to the subtleties of the different simulations used for the assessment.

202:

to “back-calculate” the approximate.. Perhaps use ‘infer’?

224-228:

See comment above. I fail to see that the structure analysis ‘largely agrees’ with the regional groupings. Perhaps rephrase this and/or discuss in more detail. I think it’s fine for the ‘inner’ group, but especially for the middle and also the outer group I am less convinced. Perhaps the PCA is clearer there, but it’s not displayed properly.

Lines 234 – 238:

“For the various MSA fishery simulations, the estimated proportions matched well with the given proportions used in the simulations (Figure S2). In the 100% simulations, the estimated proportions contained the simulated proportions for all regional reporting units apart from the outer region. In the realistic MSA simulation, the estimated proportions matched the simulated proportions well for all regional reporting units.”

This sounds all a bit vague to me. ‘matched well’, ‘estimated proportions contained the simulated proportions’ - and I am not sure how exactly these findings are informative to judge the accuracy of your baseline. What are the ‘estimated proportions’ and ‘given proportions’ here? I assume that one of the two come from your data, but which one? Presumably the ‘estimated’? Unless you manage to summarize the results from the assessment of the baseline accuracy from MSA here in a more quantitative manner, I suggest to exclude this part from the main text and move to supplementary material. As is, the result from the ‘leave-one-out’ test are much more informative.

I am not sure what Figure S2 actually shows – see comment regarding MSA for assessment of the baseline and the simulations in the methods above. There is a typo in part C of the figure – it reads ‘simmulation proportions’.

Line 241:

‘Using individuals assigned to regional units ..’ How many percent of individuals per trawl could be assigned to regional units based on your 80% probability criterion? I think this would be worthwhile reporting and I can’t seem to find these results anywhere.

Line 248-258:

I am not sure what your point is here and I suggest to work on this part. You are comparing predictions from 2008, which are based on the ecological condition of the rivers, against your results, which are also estimates (and have quite some error bars). Are you meaning to verify your results, by saying that they are close to the predictions from 2008, or are you trying to make a point that the prediction overestimated the contributions of the inner fjord populations, because they did not take into account sea-lice pressure in the fjord caused by the industry or otherwise? In any case, whatever patterns you are trying to highlight here for the inner and middle region,

they are reversed in the year 2013, which you fail to mention. So, either keep your summary more general and let the reader decide themselves based on the figure, or discuss the results more thoroughly - why do you think 2013 fell out, for example?

Line 259-261:

"While there was a trend of fish from some of the inner rivers migrating earlier than fish from the middle and outer rivers in some years, overall, there was overlap in the estimated date upon entry to saltwater among the rivers (Figure 5)."

I have already mentioned my concerns regarding these analyses above. Also, I have to say that I don't really see the trend for 'some rivers', either. The only population that comes out consistently earlier in your analyses is Opo, so if anything I would speak of one population. But, Ref 50 has found lots of individual variation in migration patterns especially for this river..

Line 261: "entry to saltwater" - Please rephrase or define clearly. where does 'saltwater' start. I guess, where the river enters the fjord. In any case, you haven't really calculated this though right? You caught fish somewhere in the fjord and calculated the time salmon spent between their rivers of origin and this place in the fjord.

265-269:

I would remove point (2) from the conclusion as a main finding - see concerns above. Also, for a main result I think it's really very vaguely put: "salmon from different rivers tended to leave their rivers around the same time".

274 - 301.

I think this part of the discussion is nice and could be kept also if you decide against retaining the analyses regarding time of departure from the rivers.

296-297:

"Sea lice prevalence tends to increase over the summer months.." Are there no more specific data available than this? Also, reference needed here. "Over the summer months" - the inner fjord populations tend to arrive a little later - that's true. But it's a matter of +/- one-two weeks. End of May at the latest, that's not yet summer, when sea lice prevalence tends to increase. The thing would have been to check for the extent of sea lice infestation in the trawls, combined with your genetic inference of river of origin..

348

I suggest 'sea-lice mediated mortality'.

350

'clearly indicate that post-smolts originating from rivers located in the inner region of the fjord face a longer migration' - I would disagree there. I don't think it's so clear.

364:

".. genetic approaches do not involve handling or tagging". I disagree there and suggest to remove this statement. In your case (genetic study) of course there was no tagging, but definitely 'handling' of fish. Unless you get your genetic data from eDNA or alike there's always going to be handling.

364-366:

For me, the last sentence is somewhat out of place. Sure, it would be good to combine all available approaches and so on to best understand what's going on. From my point of view you could already discuss telemetry/tagging studies in combination with genetic studies (your work here). Ref 50 even looked at the same rivers, but you don't put your results in context with theirs. I think, if you want to call for a multidisciplinary approach the key thing would be to combine genetic approaches, like yours presented here, with parasitological investigations, to see if fish from inner fjord populations actually show higher infestation rates with sea lice.

Figure 1:

I suggest to remove the trawl lines from the figure – it's impossible to make out specific lines - and adjust the legend accordingly. The individual trawl lines per year are also illustrated in the figures in supplementary file1 – this is enough in my opinion as far as visualization goes. Since you have the data you could also provide the start-, endpoints of the the trawl lines as supplementary data.

Line 547 – identificatino to identification.

Figure 5:

What are the points in figure 5? Are these the individual salmon that you were able to assign back to the rivers?

Supplementary file 1 –

line 22: change 'stricture' to 'structure'

Figure S1 is not displayed correctly.

Reviewer: 2

Comments to the Author(s)

Please see attached file.

Author's Response to Decision Letter for (RSOS-190426.R0)

See Appendix B.

RSOS-190426.R1 (Revision)

Review form: Reviewer 1

Is the manuscript scientifically sound in its present form?

Yes

Are the interpretations and conclusions justified by the results?

Yes

Is the language acceptable?

Yes

Do you have any ethical concerns with this paper?

No

Have you any concerns about statistical analyses in this paper?

No

Recommendation?

Accept with minor revision (please list in comments)

Comments to the Author(s)

I think the authors did a good job especially with improving the focus of the paper. The manuscript is substantially shorter now, with several parts removed, making it much more concise and readable. I only have a few comments/suggestions left - mainly of editorial nature.

Abstract:

I think the abstract fails to do justice to what for me still is the main resource of the current paper - the genetic baseline for the Hardangerfjord salmon populations. I suggest to add a few facts to the abstract, along the lines of 'established a genetic baseline based on 31 microsatellite markers for the main salmon rivers in the Hardangerfjord. We evaluated the baseline with respect to individual assignment success and MSA...'. Then it was used for MSA to evaluate the composition of trawl catches, and so on.

Line 51-53 is outdated. In the current version you limit yourself to MSA, which does not involve assigning single individuals back to rivers or regions.

Introduction:

Line 74: suggested change to: .. closely linked to the fitness and abundance ..

Line 84: suggested change to: .. genetic differentiation is low in some species ..

Line 116: suggested change to: .. is currently the world's largest producer of farmed salmon, ...

Methods:

As it is now there will be two supplementary files. The first is an excel table that contains the details on the microsats, including primers and amplification conditions across a number of tabs (never really referenced in the main manuscript as far as I can see) and the second a word document with supplementary methods (currently referred to as Supplementary file 1). I suggest to move the tables with the Microsat primers and the conditions to Supplementary file 1. This will leave the other supplementary file with just microsatellite data, which I think is a better organization of the supplementary info.

Line 184: change to something like: (for amplification conditions see Supplementary File 1)

Line 199: Provide version number of ONCOR software used for the analyses.

Line 214: The stock proportions you used for the realistic fishery simulation were obtained from the literature. Please provide the actual proportions you used as input for the realistic fishery simulation. One can sort of deduce them from fig. S2, but it would be good to provide the actual numbers.

Line: 227. As far as I can see the results from the river level MSA has been removed from the paper. I suggest to remove this sentence or present the result.

To make their analyses fully reproducible the authors need to also provide the actual stock proportions they used for the realistic fishery simulation. For the trawl microsat data which was used for the MSA I can deduce from the sample name which year it was taken, but not which week. Since the MSA was done week by week this info needs to be provided somewhere (either additional table or extra column) if the analyses should be fully reproducible.

With the extra information discussed above I think the analyses are in principle fully reproducible, so I leave the following to the editor's discretion: The Microsat data is made available as Excel tables, which is fine. I think it would help, though, if the authors would upload also the actual GENEPOP format files (Baseline file, Mixture files - would be for each week) that they used in the ONCOR analyses. I recommend that they also upload the Reporting group file

(grouping populations into regional units) and the fisheries file (parametric stock proportions for the realistic fisheries simulations).

Results:

Line 244-245: Can't assess the PCA because there are no population labels in the figure S1.

Line 248: suggested change to: .. found that on average 53.1% of fish were correctly assigned ..

Line 252: suggested change to: .. improved to 72.1% on average (Table 2).

Line 259: You have removed the analyses regarding 'date of migration from the river' and also the individual based assignment results from the paper, so please update the heading of this section.

Line 260: I find the start of this sentence confusing. Guess it must be a remnant of the previous version since all individual based assignment results were removed from the paper now. Since the MSA does not individually assign fish to a region, I suggest to remove the first part of the first sentence, and start it with: The weekly MSA..

Discussion:

- I suggest to swap the sections 'Genetic assignment success' and 'Spatio-temporal patterns of post-smolt migration'. That would seem a more logical order.

Parts of the sections 'Spatio-temporal patterns of post-smolt migration' and 'Conclusions & Practical Implications' are redundant. I suggest to either merge both sections under a new heading or, if the authors want to keep the two sections and the editor agrees, I would essentially move everything starting with the sentence 'Differences in fjord exit times' in line 284 to the 'Practical implications' section and expand/replace lines 330-337 there.

I would also move the last paragraph from the conclusions section to the start of the conclusions section. I think the section is fine, but I find it odd to close the entire manuscript with a call for combining telemetry and genetic methods, but would rather end with the call for incorporating new spatio-temporal data into existing management models, which has been really the theme of the paper under the cover..

Figure 1:

Remove the last sentence from the caption. The parts referred to have been removed from the figure.

Figure 3:

For this figure the last sentence from the caption of figure 1, i.e. the note about the trawls not going over land, would be appropriate.

Supplementary File 1:

Please add labels (river) to the PCA (Fig. S1). I just see blue dots without labels.

Review form: Reviewer 2

Is the manuscript scientifically sound in its present form?

Yes

Are the interpretations and conclusions justified by the results?

Yes

Is the language acceptable?

Yes

Do you have any ethical concerns with this paper?

No

Have you any concerns about statistical analyses in this paper?

No

Recommendation?

Accept with minor revision (please list in comments)

Comments to the Author(s)

See attached document (Appendix C).

Decision letter (RSOS-190426.R1)

13-Aug-2019

Dear Dr Harvey:

On behalf of the Editors, I am pleased to inform you that your Manuscript RSOS-190426.R1 entitled "Inferring Atlantic salmon post-smolt migration patterns using genetic assignment" has been accepted for publication in Royal Society Open Science subject to minor revision in accordance with the referee suggestions. Please find the referees' comments at the end of this email.

The reviewers and Subject Editor have recommended publication, but also suggest some minor revisions to your manuscript. Therefore, I invite you to respond to the comments and revise your manuscript.

- Ethics statement

- Data accessibility

<http://datadryad.org/submit?journalID=RSOS&manu=RSOS-190426.R1>

- **Competing interests**

- **Authors' contributions**

- **Acknowledgements**

- **Funding statement**

Because the schedule for publication is very tight, it is a condition of publication that you submit the revised version of your manuscript before 22-Aug-2019. Please note that the revision deadline will expire at 00.00am on this date. If you do not think you will be able to meet this date please let me know immediately.

on behalf of Dr Kristina Sefc (Associate Editor) and Kevin Padian (Subject Editor)
openscience@royalsociety.org

Associate Editor Comments to Author (Dr Kristina Sefc):

Associate Editor: 1

Comments to the Author:

Both reviewers commend the authors for the good job done with the revision. The reviews identify information that still needs to be added to the manuscript for clarity and offer suggestions for improved wording and structuring. In addition to following these suggestions, please make sure to provide the information on sampling week and the two files (reporting group and fisheries) requested by Reviewer 1. Also, please make sure that the excel table with the microsatellite data contains all necessary information to convert it into the genepop files used in your analyses, or provide the genepop files.

Reviewer comments to Author:

Reviewer: 1

Comments to the Author(s)

I think the authors did a good job especially with improving the focus of the paper. The manuscript is substantially shorter now, with several parts removed, making it much more concise and readable. I only have a few comments/suggestions left - mainly of editorial nature.

Abstract:

I think the abstract fails to do justice to what for me still is the main resource of the current paper - the genetic baseline for the Hardangerfjord salmon populations. I suggest to add a few facts to the abstract, along the lines of 'established a genetic baseline based on 31 microsatellite markers for the main salmon rivers in the Hardangerfjord. We evaluated the baseline with respect to individual assignment success and MSA...'.. Then it was used for MSA to evaluate the composition of trawl catches, and so on.

Line 51-53 is outdated. In the current version you limit yourself to MSA, which does not involve assigning single individuals back to rivers or regions.

Introduction:

Line 74: suggested change to: .. closely linked to the fitness and abundance ..

Line 84: suggested change to: .. genetic differentiation is low in some species ..

Line 116: suggested change to: .. is currently the world's largest producer of farmed salmon, ...

Methods:

As it is now there will be two supplementary files. The first is an excel table that contains the details on the microsats, including primers and amplification conditions across a number of tabs (never really referenced in the main manuscript as far as I can see) and the second a word document with supplementary methods (currently referred to as Supplementary file 1). I suggest to move the tables with the Microsat primers and the conditions to Supplementary file 1. This will leave the other supplementary file with just microsatellite data, which I think is a better organization of the supplementary info.

Line 184: change to something like: (for amplification conditions see Supplementary File 1)

Line 199: Provide version number of ONCOR software used for the analyses.

Line 214: The stock proportions you used for the realistic fishery simulation were obtained from the literature. Please provide the actual proportions you used as input for the realistic fishery simulation. One can sort of deduce them from fig. S2, but it would be good to provide the actual numbers.

Line: 227. As far as I can see the results from the river level MSA has been removed from the paper. I suggest to remove this sentence or present the result.

To make their analyses fully reproducible the authors need to also provide the actual stock proportions they used for the realistic fishery simulation. For the trawl microsat data which was used for the MSA I can deduce from the sample name which year it was taken, but not which week. Since the MSA was done week by week this info needs to be provided somewhere (either additional table or extra column) if the analyses should be fully reproducible.

With the extra information discussed above I think the analyses are in principle fully reproducible, so I leave the following to the editor's discretion: The Microsat data is made available as Excel tables, which is fine. I think it would help, though, if the authors would upload also the actual GENPOP format files (Baseline file, Mixture files - would be for each week) that they used in the ONCOR analyses. I recommend that they also upload the Reporting group file (grouping populations into regional units) and the fisheries file (parametric stock proportions for the realistic fisheries simulations).

Results:

Line 244-245: Can't assess the PCA because there are no population labels in the figure S1.

Line 248: suggested change to: .. found that on average 53.1% of fish were correctly assigned ..

Line 252: suggested change to: .. improved to 72.1% on average (Table 2).

Line 259: You have removed the analyses regarding 'date of migration from the river' and also the individual based assignment results from the paper, so please update the heading of this section.

Line 260: I find the start of this sentence confusing. Guess it must be a remnant of the previous version since all individual based assignment results were removed from the paper now. Since the MSA does not individually assign fish to a region, I suggest to remove the first part of the first sentence, and start it with: The weekly MSA..

Discussion:

- I suggest to swap the sections 'Genetic assignment success' and 'Spatio-temporal patterns of post-smolt migration'. That would seem a more logical order.

Parts of the sections 'Spatio-temporal patterns of post-smolt migration' and 'Conclusions & Practical Implications' are redundant. I suggest to either merge both sections under a new heading or, if the authors want to keep the two sections and the editor agrees, I would essentially move everything starting with the sentence 'Differences in fjord exit times' in line 284 to the 'Practical implications' section and expand/replace lines 330-337 there.

I would also move the last paragraph from the conclusions section to the start of the conclusions section. I think the section is fine, but I find it odd to close the entire manuscript with a call for combining telemetry and genetic methods, but would rather end with the call for incorporating new spatio-temporal data into existing management models, which has been really the theme of the paper under the cover..

Figure 1:

Remove the last sentence from the caption. The parts referred to have been removed from the figure.

Figure 3:

For this figure the last sentence from the caption of figure 1, i.e. the note about the trawls not going over land, would be appropriate.

Supplementary File 1:

Please add labels (river) to the PCA (Fig. S1). I just see blue dots without labels.

Reviewer: 2**Comments to the Author(s)**

See attached document.

Author's Response to Decision Letter for (RSOS-190426.R1)

See Appendix D.

Decision letter (RSOS-190426.R2)

20-Aug-2019

Dear Dr Harvey,

I am pleased to inform you that your manuscript entitled "Inferring Atlantic salmon post-smolt migration patterns using genetic assignment" is now accepted for publication in Royal Society Open Science.

on behalf of Dr Kristina Sefc (Associate Editor) and Kevin Padian (Subject Editor)
openscience@royalsociety.org

Appendix A

Review of RSOS, Atlantic post smolt migr using GSI.

General

This manuscript investigates how genetic stock identification (mixed-stock analyses (MSA) and individual assignment) can be used to determine river or regional origin of Atlantic salmon post-smolts, and how this can be used to determine spatial and temporal migration patterns of out-of-fjord migration. There is an underlying goal (not explicitly stated) to be able to use the results in a migration model with the purpose to inform managers in the protective work with sea-lice infestation in relation to salmon fish farms in Norway. They use a genetic baseline of salmon rivers (populations) in the fjord and data from trawl fishing in the outer fjord, which captures post smolts of salmon. They show that there is not satisfactory assignment power to determine river of origin of individual fish, but using regional reporting groups the assignment works well. The results also show that the salmon from inner-fjord rivers use longer time in their migration and therefore are more sensible to sea lice infestation than salmon from the middle- and outer-fjord rivers.

The topic is interesting and highly relevant for managers dealing with salmon aquaculture. Maybe not so much to a more general audience, however the salmon aquaculture is an enormous business affecting most people as many eat salmon several times per year, and most of the salmon comes from Norway.

The idea to use GSI to identify post-smolts caught in trawls, and potentially use the information in other models, is very nice. The authors claim this being the first study to infer fjord migration of wild Atlantic post-smolts using genetic methods. This might be true, but is far from a novel idea. GSI have been used in many contexts and this is just one application, even though interesting.

Unfortunately, the manuscript have several weaknesses. My main concerns is the actual outline and aims of the paper, which to me is confusing. I find it to have too much focus on the sea lice problem and aquaculture, while the actual research question investigated is genetic stock assignment methods and how these can be used to determine migration timing and patterns. I understand that the authors wants to emphasise the applications in “the next step”, but this manuscript does not use any information in any new model, e.g. estimating actual infestation rates or development of a new model where migration information and genetic data is combined (see for example Whitlock et al., 2018). They simply use GSI to investigate timing and migration patterns of different salmon populations.

An example of the above is the last sentence in the abstract that “these results are directly implemented in the models...” which is simply not true, at least this is not done in this manuscript.

A second concern is that I believe the authors have used results of ONCOR a bit too naively. They have used the self-assignment and 100% simulation, but should have also used some empirical data of known origin to provide assessment of the potential accuracy of their baseline to perform GSI.

In addition, even though the GSI is the main actual aim of the manuscript, the introduction and part of conclusion of GSI is taken too lightly, meaning that it is taken for granted that the reader knows what GSI is, how it works, and potential problems with a genetic baseline and assignment procedures is ignored. I provide more detailed comments in this matter below.

In summary, I am not convinced that this manuscript fits the general scope of RSOS, may be should better fit in a fisheries (biology) journal. It is clear though, that it needs re-working quite a bit.

Specific

Throughout the manuscript there is an inconsistency in writing numbers with letters or numbers (e.g. four or 4) check that!

Line 56-58. "...will be at higher risk..." likely, but you didn't test that in this manuscript.

Line 58-60. You did not implement anything in any model in this manuscript. I don't see at all why you can claim this here??

Line 73-103. Too much focus on the aquaculture and sea lice problem for a manuscript with GSI as main aim.

Line 104-107. There is no info here on what GSI is or how it does work. If the aim is a more general public, more information on the methods, its pros and cons, should be mentioned.

Line 110. In line with above, any good GSI study requires adequate baseline samples, but it is not only "fine-scale". There are several factors contributing to a successful genetic assignment, e.g. nr of markers, quality of markers, nr of alleles, genetic differentiation among baseline samples, representative sampling etc. The whole section in the introduction of GSI is very weak, especially since the main aim of the manuscript is GSI.

Line 118-120. Here you state the aims of the manuscript. There is nothing mentioned about the link to new models for management of sea lice problems in relation to aquaculture. Fine, but you later and in the abstract claim there is??

Line 124-149 This section is not well designed. It is not easy to follow. Try so re-order parts and get it more focused. Maybe with a first paragraph with the overall design, or start with the baseline samples, which is a more logical order. You need the baseline to be able to assign trawl data.

Line 141 sentence doens not make sense.

Line 143. OK, but how does this with different life stages affect your baseline?

Line 146. I don't understand. Comparison between farmed and wild is not mentioned before??

Line 148. Assignment of what?

Line 156. I don't know if you just can write "available on request". It makes it impossible to review.

Line 159. Quality checked??? How??? What was the results???????

Line 164. Again, I don't know if the journal allows this "standard methods". I don't think a reader should have to go through a supplement to understand what methods was used.

Line 168-182. The LOO test and 100%-simulations have a problem that it may over-estimate accuracy.

Line 190. Not very impressive with visual examination. Better to do statistical tests.

Line 196. The cut-off level of 0.80. What was that based on? How many individuals was then removed? Or how large proportion?

Line 197-205. This section is not easy to follow. Try to make more clear.

Line 208-121. Ok, but there are studies that show a slower migration speed in reared salmon and sea trout than in wild counterparts, despite the larger size.

In the results section, please write Supporting information when the tables belong there, not just e.g. Table S1.

Line 219. And?

Line 220. Add reference to Evanno.

IN the results, some times MSA is used and sometimes mixed stock analyses, be consistent.

Line 245-247. Do statistical tests! Or what does trends mean??

Line 264. What patterns?

Line 269. The conclusion here is not really surprising.

Line 271. Remove the part of sentence after the ,

Line 275-283 This is an odd section, kind of an introduction. Maybe it would fit better there, instead of the too big focus on aquaculture.

Line 305. What post-smolt migration model? You cannot just put a reference.

Line 309. You are speculating.

Line 314-316. I don't think you can draw the conclusion that just because there was evidence of genetic structure, the region was better assignment that river. There are so many more factors affecting assignment success in GSI. Some mentioned earlier.

Line 317. Good balance? Please explain. This does not make sense.

Line 319-327 This is something that was not mentioned in the introduction, nor in the methods, i.e. that you would do an estimation of smolt production. This I would omit completely. It is just confusing, and a correct assignment would need a well planned randomized trawl sampling design to start with.

Line 328-330. Please see Whitlock et al. 2018 for an example of this.

Line 334. This was not mentioned before, i.e. in the methods for accuracy evaluation. I don't really see how this can provide any valuable information on accuracy, since you don't know the true origin of assigned fish.

Line 364-365. There are a few studies that have combined genetics and telemetry, e.g. Östergren et al., 2012.

Figures

Figure 1. what is geographic assignment units? In the ms you use reporting units.

Figure 2. Is letters A and B missing to show which is which? i.e. above and below.

Figure 3. Way too small text.

Figure 4. Not sure that reference is need in the figure text.

Figure 5. What are the dots??

Table 1. A bit confusing table. What is different between fins and fin clip?

References

Whitlock, R., Mäntyniemi, S., Palm, S., Koljonen, M.-L., Dannewitz, J., Östergren, J. (2018) Integrating genetic analysis of mixed populations with a spatially-explicit population dynamics model. *Methods in Ecology and Evolution*. 2018:1-19 DOI: 10.1111/2041-210X.12946.

Östergren, J., Nilsson, J. & Lundqvist, H. (2012). Linking genetic assignment tests with telemetry enhances understanding of spawning migration and homing in sea trout *Salmo trutta* L. *Hydrobiologia*. (DOI) 10.1007/s10750-012-1063-7.

Appendix B

Review received: 7 May 2019

Re-submitted: 21 June 2019

Response to reviewer comments RSOS-190426

We greatly appreciate the editors and reviewers taking the time to go through the paper again and provide further suggestions and feedback. We have carefully answered the editor and reviewer comments, and our responses are provided below.

Associate Editor's comments (Dr Kristina Sefc):

The manuscript has now been seen by two reviewers. Both highlight the value of this impressive dataset, but both are also quite critical about some aspects of the study. In particular, the reviewers emphasize the need to adapt the manuscript to a general (non-fisheries) readership. Please follow their advice, i.e. distinguish between the actual study (genetic assignment of post-smolts) and possible applications, and in connection with the latter, provide the readers with more information about the salmon industry.

- *The introduction has now been reworked to incorporate all reviewer comments where applicable. Please see below for more details.*

Furthermore, please follow the reviewers' suggestions regarding analyses and interpretation of the data. If you decide to retain the analyses of migration times in the manuscript (see reviewer 1), please make sure to discuss potential uncertainties and caveats.

- *The analyses have now been changed in accordance with the reviewer suggestions. Please see below for more details.*

Additionally, please take care to provide the requested information about the microsatellite loci / genotyping.

- *This information is now contained in Supplementary File S2.*

From my own reading of the manuscript:

Table 1: why is number of samples often lower than number genotyped?

- *Sometimes some samples do not genotype correctly, or it turns out that the fish are trout and not salmon, which is detected in the genotyping, so often the number of samples genotyped are higher than the number of genotyped samples then used in further analyses*

Table 1: Is sample type in any way important with regard to the study? If you retain the column, don't use a sample type category "DNA" – all of the other samples (fins, scales etc.) were used to extract DNA, and some tissue must have been collected from the Etneelva fish for DNA extraction as well.

- *The column has been removed (see Table 1)*

In order to determine whether the Structure analysis indeed supports the regional assignments (inner/middle/outer), you should investigate the results for different values of K. With $K < 5$, does the assignment approach a reflection of the three regions?

- *The Structure analysis was run for all K values ($K = 12$, using only the baseline without the 2 outlier populations), and with $K = 3$ and 4 there was a distinction between the inner populations and the middle & outer populations. As suggested by reviewer 1, we have amended the text relating to the regional choices to show that Structure agreed to an extent, while geography was the main reason behind the regional assignment approach (lines 239 – 243).*

line 103: sea lice induced mortality

- *This has been changed (line 125)*

Fig. 5 lacks a color legend for the inner/middle/outer color codes.

- *This figure has now been removed at the request of the reviewers (see below)*

The legend of Fig. 4 is quite challenging.

- *This figure has now been removed at the request of the reviewers (see below)*

Comments to Author:

Reviewers' Comments to Author:

Reviewer: 1

Comments to the Author(s)

Harvey et al. report on post-smolt migration patterns of Atlantic salmon in the Hardangerfjord in Norway. The authors present a genetic baseline for the major Atlantic salmon populations in the Hardangerfjord (based on 31 microsatellites) and attempt to assign post smolt salmon caught in the fjord (4 years worth of data) back to their native rivers/regions. Their data indicate that salmon from rivers draining into the inner fjord regions arrive at the outer reaches of the fjord later, compared to populations from middle and outer fjord river populations. They further investigate the timing of migration of the populations and discuss the implications of their findings with respect to current regulations in salmon production.

I congratulate the authors to a well written manuscript. The size of the dataset (especially for the genetic baseline) is quite impressive, the paper is interesting and I think the analyses are generally sound (but see below). In my opinion, the key resources presented by the paper are the genetic baseline for the major Atlantic salmon populations in the Hardangerfjord, the evaluation of assignment success of post-smolts to their rivers/regions of origin, and the results regarding the arrival times of the populations in the outer fjord region.

I fully agree that more research is needed and every effort should be made to refine regulations for salmon production in the fjord to reduce the effect of industry-mediated

parasite pressure on wild salmon smolts. However, the inference regarding migration times as well as the discussion of the implications of these are in my opinion a bit superficial. The authors adopt a migration rate from the literature and calculate the time of departure of salmon based on this and the shortest distance (actually it's not even clear to me which distance they chose) between the river of origin and the location where the fish were caught. I think they should either be done/discussed more thoroughly or removed from the manuscript. I do think this manuscript would be fine without these parts given the impressive amount of data it presents. Halttunen et al. 2018 (ref. 50 in current study) found that individual variability in progression rate and route choice affected by hydrographic conditions had strong effects on fjord residency in Atlantic salmon. Additionally the study found that most fish did not swim directly to the outer straights.

- *These analyses have now been removed. Please see below for details*

I also recommend that the authors increase their efforts with respect to the reproducibility of their analyses, i.e. sharing of data - see below.

- *This has been done. Please see below.*

The journal targets a general readership, so mainly people outside the Norwegian salmon production industry. Therefore, I would recommend that the authors give a bit more detail about the current regulations and the models, which were used to inform those. As it is now, the authors report that there are models that predict temporal and spatial overlap between sea lice larvae and post-smolts, and that these are used to 'regulate the production level of the aquaculture industry'. I think for a general readership it would be interesting to know what that means in practice. Do certain production sites in certain parts of the fjord have to be shut down completely over a period, or reduce the number of fish produced, currently when, etc.? Perhaps the authors could even give an example of a production site for which the current regulations dictate that it is shut down in the period from month A to month B and discuss how this strategy is at odds with their findings.

- *This has been expanded on (lines 120-129)*

The extent the authors decide to elaborate on this further may of course depend on how much of the focus on the implications for the regulations they decide to retain in a revised manuscript.

Other general comments:

The authors have chosen the colors green, blue, red to represent the outer, middle and inner fjord regions. I suggest to consider readers with red-green color blindness and change this color scheme.

- *All figures have now been changed*

Reproducibility:

- line 155: The analyses are based on 31 microsatellites. The authors provide essentially no information on them except for the IDs in the supplementary table 2. To me it's not clear if

they were published previously or are new, whether the authors present the multiplexing assay for the first time, etc. I don't think providing information on conditions on request is sufficient and suggest that the authors provide details (references for published microsatellites, primer sequence in case of new microsatellites, PCR/multiplexing conditions, etc.) in a supplementary table/document, if not in the main text.

- *Primer information (sequences and references) and PCR protocols have now been added to supplementary file 2*

- In supplementary File S2, baseline regions tab – please add a column with river name to the table

- *This has been added, please see the new supplementary File S2*

- In supplementary File S2, trawl all years tab – please add a column with the assignment results for the individuals, e.g. river/region assigned to, probability, unassigned, etc.

- *The individual assignments are not used in this study now (analyses pertaining to individual assignment have been removed). The MSA output does not individually assign fish to a region, it presents an estimate (with Cis) of the proportion of fish in each region as a percentage value, therefore the regional analysis was not done to an individual level. The MSA outputs for each year are attached in the Supplementary File S2.*

- If you decide to retain this part, please provide the minimum distance by sea between the capture point and river of origin you used to calculate the migration times in a supplementary table.

- *These analyses have now been removed.*

Comments by line, typos, suggestions:

Lines 58-60:

Last sentence of the abstract: 'These results are directly implemented ..' When I first read the abstract I was expecting that the current study presents a recalibrated model, which is not the case. I suggest to remove this sentence or rephrase it towards 'the new data should be incorporated into the existing models'.

- *The line has now been changed to reflect the intended meaning: that these results will be (are being) used within the models that estimate additional sea lice mortality from aquaculture (line 60)*

114:

we developed an genetic
should be 'developed a genetic'

- *This has been changed (line 139)*

130:

which anaesthetic?

- *The name of the anesthetic has been added (line 157)*

132:

I suggest to start a new sentence here: ‘The 2014 trawl samples were not included ..

- *This has been changed (line 159)*

148:

“Assignment analyses were carried out..” this sentence should be removed from the section about establishing the genetic baseline towards the next section (around line 169).

- *The sentence has been moved as suggested (line 197)*

155:

In total, 31 microsatellite markers were amplified in five PCR multiplexes (amplification conditions are available on request).

Need to provide references, primers and conditions as supplementary table/document. See comment on reproducibility.

- *This has now been done (Supplementary file S2)*

168:

“STRUCTURE and PCA were used to ... and identify potential regional reporting units.. “. I suggest to remove/change this sentence. It seems to me that the regional reporting units are defined mainly based on geographical location in the fjord. I am not saying that these as defined here are necessarily inappropriate, the self-assignment test based on these certainly shows that – it just doesn’t seem to me that it’s fair to say the regional reporting units are based on the microsat data. Looking at the structure plot, especially the middle reporting unit is quite a mix and I wonder if an objective clustering criterion (without a priori geographical information) would group the populations together. Perhaps the PCA shows a clearer pattern, but it is not displayed properly in the supporting file, so I can’t assess that.

- *The sentence has been changed so that only the analysis of genetic structure is referred to (line 196)*

Lines 176-182:

The ‘leave-one-out’ test is reasonably straight forward, but I have to admit that it’s not entirely clear to me how the MSA works for assessing the accuracy of the baseline. I assume that the general reader won’t be either, so I’d suggest that the authors extend this part slightly, also with respect to the subtleties of the different simulations used for the assessment.

- *This section has now been reworked to include more detail on the MSA assessment (lines 204-217)*

202:

to “back-calculate” the approximate.. Perhaps use ‘infer’?

- *These analyses have been removed*

224-228:

See comment above. I fail to see that the structure analysis ‘largely agrees’ with the regional groupings. Perhaps rephrase this and/or discuss in more detail. I think it’s fine for the ‘inner’ group, but especially for the middle and also the outer group I am less convinced. Perhaps the PCA is clearer there, but it’s not displayed properly.

- *The section has been changed. Now the structure results are discussed and only the clustering of the inner rivers are referred to in relation to the geographic regional assignment. The patterns of the middle and outer rivers are discussed separately (lines 239 -243).*

Lines 234 – 238:

“For the various MSA fishery simulations, the estimated proportions matched well with the given proportions used in the simulations (Figure S2). In the 100% simulations, the estimated proportions contained the simulated proportions for all regional reporting units apart from the outer region. In the realistic MSA simulation, the estimated proportions matched the simulated proportions well for all regional reporting units.”

This sounds all a bit vague to me. ‘matched well’, ‘estimated proportions contained the simulated proportions’ - and I am not sure how exactly these findings are informative to judge the accuracy of your baseline. What are the ‘estimated proportions’ and ‘given proportions’ here? I assume that one of the two come from your data, but which one? Presumably the ‘estimated’? Unless you manage to summarize the results from the assessment of the baseline accuracy from MSA here in a more quantitative manner, I suggest to exclude this part from the main text and move to supplementary material. As is, the result from the ‘leave-one-out’ test are much more informative.

- *This section has been reworded (lines 251 -256)*

I am not sure what Figure S2 actually shows – see comment regarding MSA for assessment of the baseline and the simulations in the methods above. There is a typo in part C of the figure – it reads ‘simmulation proportions’.

- *This has been corrected (Figure S2)*

Line 241:

‘Using individuals assigned to regional units ..’ How many percent of individuals per trawl could be assigned to regional units based on your 80% probability criterion? I think this would be worthwhile reporting and I can’t seem to find these results anywhere.

- *These assignments were done by a mixture analysis, where the results are given as proportions of each baseline population in the mixture (the trawl per week) and so the fish are not individually assigned, therefore there was no cut off, that was only used for the individual assignment, which was used to estimate the timing of migration into the fjord, which has now been removed.*

-

Line 248-258:

I am not sure what your point is here and I suggest to work on this part. You are comparing predictions from 2008, which are based on the ecological condition of the rivers, against your

results, which are also estimates (and have quite some error bars). Are you meaning to verify your results, by saying that they are close to the predictions from 2008, or are you trying to make a point that the prediction overestimated the contributions of the inner fjord populations, because they did not take into account sea-lice pressure in the fjord caused by the industry or otherwise? In any case, whatever patterns you are trying to highlight here for the inner and middle region, they are reversed in the year 2013, which you fail to mention. So, either keep your summary more general and let the reader decide themselves based on the figure, or discuss the results more thoroughly - why do you think 2013 fell out, for example?

- *This section has now been removed at the suggestion of reviewer 2.*

Line 259-261:

“While there was a trend of fish from some of the inner rivers migrating earlier than fish from the middle and outer rivers in some years, overall, there was overlap in the estimated date upon entry to saltwater among the rivers (Figure 5).”

I have already mentioned my concerns regarding these analyses above. Also, I have to say that I don't really see the trend for 'some rivers', either. The only population that comes out consistently earlier in your analyses is Opo, so if anything I would speak of one population. But, Ref 50 has found lots of individual variation in migration patterns especially for this river..

- *This section has been removed as suggested*

Line 261: “entry to saltwater” – Please rephrase or define clearly. where does ‘saltwater’ start. I guess, where the river enters the fjord. In any case, you haven't really calculated this though right? You caught fish somewhere in the fjord and calculated the time salmon spent between their rivers of origin and this place in the fjord.

- *This section has been removed as suggested*

265-269:

I would remove point (2) from the conclusion as a main finding – see concerns above. Also, for a main result I think it's really very vaguely put: “salmon from different rivers tended to leave their rivers around the same time “.

- *This has been removed*

274 – 301.

I think this part of the discussion is nice and could be kept also if you decide against retaining the analyses regarding time of departure from the rivers.

- *This section has been moved to the introduction per suggestion by reviewer 2*

296-297:

“Sea lice prevalence tends to increase over the summer months..” Are there no more specific data available than this? Also, reference needed here. “Over the summer months” – the inner fjord populations tend to arrive a little later – that's true. But it's a matter of +- one-two weeks. End of May at the latest, that's not yet summer, when sea lice prevalence tends to

increase. The thing would have been to check for the extent of sea lice infestation in the trawls, combined with your genetic inference of river of origin..

- *This section has now been reworded and an appropriate reference added. (line 289)*

348

I suggest 'sea-lice mediated mortality'.

- *This has been changed as suggested (line 327)*

350

'clearly indicate that post-smolts originating from rivers located in the inner region of the fjord face a longer migration' – I would disagree there. I don't think it's so clear.

- *The word "clearly" has been removed to tone down the meaning.*

364:

".. genetic approaches do not involve handling or tagging". I disagree there and suggest to remove this statement. In your case (genetic study) of course there was no tagging, but definitely 'handling' of fish. Unless you get your genetic data from eDNA or alike there's always going to be handling.

- *This has been changed (line 343)*

364-366:

For me, the last sentence is somewhat out of place. Sure, it would be good to combine all available approaches and so on to best understand what's going on. From my point of view you could already discuss telemetry/tagging studies in combination with genetic studies (your work here). Ref 50 even looked at the same rivers, but you don't put your results in context with theirs. I think, if you want to call for a multidisciplinary approach the key thing would be to combine genetic approaches, like yours presented here, with parasitological investigations, to see if fish from inner fjord populations actually show higher infestation rates with sea lice.

- *Telemetry is now introduced earlier (79-85) and studies using both methods are mentioned. As mentioned in the introduction, the results from an individual assignment analysis are being used with the sea lice data in another study (Johnsen et al, submitted).*

Figure 1:

I suggest to remove the trawl lines from the figure – it's impossible to make out specific lines - and adjust the legend accordingly. The individual trawl lines per year are also illustrated in the figures in supplementary file1 – this is enough in my opinion as far as visualization goes. Since you have the data you could also provide the start-, endpoints of the the trawl lines as supplementary data.

- *The trawl lines have now been removed from the figure*

Line 547 – identificatino to identification.

- *This has been changed*

Figure 5:

What are the points in figure 5? Are these the individual salmon that you were able to assign back to the rivers?

- *This figure has now been removed*

Supplementary file 1 –

line 22: change ‘stricture’ to ‘structure’

- *This has been changed (line 22)*

Figure S1 is not displayed correctly.

- *There were no obvious display issues with Figure S1.*

Reviewer: 2

General

This manuscript investigates how genetic stock identification (mixed-stock analyses (MSA) and individual assignment) can be used to determine river or regional origin of Atlantic salmon post-smolts, and how this can be used to determine spatial and temporal migration patterns of out-of-fjord migration. There is an underlying goal (not explicitly stated) to be able to use the results in a migration model with the purpose to inform managers in the protective work with sea-lice infestation in relation to salmon fish farms in Norway. They use a genetic baseline of salmon rivers (populations) in the fjord and data from trawl fishing in the outer fjord, which captures post smolts of salmon. They show that there is not satisfactory assignment power to determine river of origin of individual fish, but using regional reporting groups the assignment works well. The results also show that the salmon from inner-fjord rivers use longer time in their migration and therefore are more sensible to sea lice infestation than salmon from the middle- and outer-fjord rivers.

The topic is interesting and highly relevant for managers dealing with salmon aquaculture. Maybe not so much to a more general audience, however the salmon aquaculture is an enormous business affecting most people as many eat salmon several times per year, and most of the salmon comes from Norway.

The idea to use GSI to identify post-smolts caught in trawls, and potentially use the information in other models, is very nice. The authors claim this being the first study to infer fjord migration of wild Atlantic post-smolts using genetic methods. This might be true, but is far from a novel idea. GSI have been used in many contexts and this is just one application, even though interesting.

Unfortunately, the manuscript have several weaknesses. My main concerns is the actual outline and aims of the paper, which to me is confusing. I find it to have too much focus on the sea lice problem and aquaculture, while the actual research question investigated is genetic stock assignment methods and how these can be used to determine migration timing and patterns. I understand that the authors wants to emphasise the applications in “the next step”, but this manuscript does not use any information in any new model, e.g. estimating actual infestation rates or development of a new model where migration information and genetic data is combined (see for example Whitlock et al., 2018). They simply use GSI to investigate timing and migration patterns of different salmon populations. An example of the above is the last sentence in the abstract that “these results are directly implemented in the models...” which is simply not true, at least this is not done in this manuscript.

- *The aims have now been clarified (line 143-147) with regard to their application in the present study and in another study, where results from the individual assignment (no longer a focus of this MS) are used.*

A second concern is that I believe the authors have used results of ONCOR a bit too naively. They have used the self-assignment and 100% simulation, but should have also used some empirical data of known origin to provide assessment of the potential accuracy of their baseline to perform GSI. In addition, even though the GSI is the main actual aim of the manuscript, the introduction and part of conclusion of GSI is taken too lightly, meaning that it is taken for granted that the reader knows what GSI is, how it works, and potential problems with a genetic baseline and assignment procedures is ignored. I provide more detailed comments in this matter below.

- *Along with the 100% simulations we also used realistic fishery simulations which take fish from the baseline and assign them back to the truncated baseline (209-215). Further, when estimating the 0.80 probability cut off (which has now been removed from the MS) we also removed fish randomly from the baseline and assigned them back to the truncated baseline. So the accuracy analysis was based on both simulations and empirical*

In summary, I am not convinced that this manuscript fits the general scope of RSOS, may be should better fit in a fisheries (biology) journal. It is clear though, that it needs re-working quite a bit.

Specific

Throughout the manuscript there is an inconsistency in writing numbers with letters or numbers (e.g. four or 4) check that!

- *This has now been changed (Line 101, 164, 233, 282)*

-

Line 56-58. "...will be at higher risk..." likely, but you didn't test that in this manuscript.

- *The sentence has been changed to reflect the intended meaning (line 58)*

Line 58-60. You did not implement anything in any model in this manuscript. I don't see at all why you can claim this here??

- *The sentence has been amended to reflect the intended meaning (line 60)*

Line 73-103. Too much focus on the aquaculture and sea lice problem for a manuscript with GSI as main aim.

- *The introduction has been reworked according to the suggestions by all reviewers and the editor, it has been broadened to suit a more general readership with a focus on the aquaculture industry and its regulation, and information on GSI (lines 73-95, 120-129)*

Line 104-107. There is no info here on what GSI is or how it does work. If the aim is a more general public, more information on the methods, its pros and cons, should be mentioned.

- *GSI is expanded on in the introduction in line with this comment (lines 88-93)*

Line 110. In line with above, any good GSI study requires adequate baseline samples, but it is not only "fine-scale". There are several factors contributing to a successful genetic assignment, e.g. nr of markers, quality of markers, nr of alleles, genetic differentiation among baseline samples,

Line 124-149 This section is not well designed. It is not easy to follow. Try so re-order parts and get it more focused. Maybe with a first paragraph with the overall design, or start with the baseline samples, which is a more logical order. You need the baseline to be able to assign trawl data.

- *These sections have now been switched.*

Line 141 sentence does not make sense.

- *The sentence has been amended (line 159)*

Line 143. OK, but how does this with different life stages affect your baseline?

- *This line has been removed*

Line 146. I don't understand. Comparison between farmed and wild is not mentioned before??

- *The sentence has been amended (line 163)*

Line 148. Assignment of what?

- *This sentence has been moved to later in the MS where it fits better (line 197)*

Line 156. I don't know if you just can write "available on request". It makes it impossible to review.

- *The relevant information (microsatellite references and PCR protocols) has now been included in the Supplementary File 2.*

Line 159. Quality checked??? How??? What was the results???????

- *Quality checked by another person to ensure that the microsatellite scoring was correct and in line with the laboratory standards. (line 178-179)*

Line 164. Again, I don't know if the journal allows this "standard methods". I don't think a reader should have to go through a supplement to understand what methods was used.

- *This was done to conserve space in the manuscript.*

Line 168-182. The LOO test and 100%-simulations have a problem that it may over-estimate accuracy.

- *As above, realistic fishery simulations were also carried out (see above)*

Line 190. Not very impressive with visual examination. Better to do statistical tests.

- *It was felt that statistical tests of the present data (proportional MSA estimates by week) was not appropriate, and that the patterns of regional MSA are clear. The low numbers of data (4 estimates (one per region) per week (four weeks each year) would introduce high uncertainty. Confidence intervals are included to show the estimated uncertainty around the estimated proportions.*

Line 196. The cut-off level of 0.80. What was that based on? How many individuals was then removed? Or how large proportion?

- *This analysis (the individual assignment was used only for calculating migration timing) has now been removed. The 0.80 threshold level was based on using test sets of “unknown” individuals removed from the baseline before assignment and then assigned back. The threshold was taken as the probability value that represents a high level of correctly assigned individuals and overall assigned individuals*

Line 197-205. This section is not easy to follow. Try to make more clear.

- *This section has been removed in line with editor and reviewer comments*

Line 208-121. Ok, but there are studies that show a slower migration speed in reared salmon and sea trout than in wild counterparts, despite the larger size.

- *This section has been removed in line with editor and reviewer comments*

In the results section, please write Supporting information when the tables belong there, not just e.g. Table S1.

- *This has been added (line 232, 244, 248, 254, 256,265)*

Line 219. And?

- *As the analyses were taken to the regional level, this difference between the years was no longer of interest.*

Line 220. Add reference to Evanno.

- *The reference has been added (line 236)*

IN the results, some times MSA is used and sometimes mixed stock analyses, be consistent.

- *This has been changed where applicable*

Line 245-247. Do statistical tests! Or what does trends mean??

- *See above*

Line 264. What patterns?

- *This has been changed (line 268)*

Line 269. The conclusion here is not really surprising.

- *This part has been removed*

Line 271. Remove the part of sentence after the ,

- *This has been removed*

Line 275-283 This is an odd section, kind of an introduction. Maybe it would fit better there, instead of the too big focus on aquaculture.

- *This section has now been incorporated into the introduction*

Line 305. What post-smolt migration model? You cannot just put a reference.

- *This section has been removed*

Line 309. You are speculating.

- *This section has been removed*

Line 314-316. I don't think you can draw the conclusion that just because there was evidence of genetic structure, the region was better assignment than river. There are so many more factors affecting assignment success in GSI. Some mentioned earlier.

- *This sentence has been changed (line 298)*

Line 317. Good balance? Please explain. This does not make sense.

- *The sentence has been changed (line 300)*

Line 319-327 This is something that was not mentioned in the introduction, nor in the methods, i.e. that you would do an estimation of smolt production. This I would omit completely. It is just confusing, and a correct assignment would need a well planned randomized trawl sampling design to start with.

- *This section has been removed*

Line 328-330. Please see Whitlock et al. 2018 for an example of this.

Line 334. This was not mentioned before, i.e. in the methods for accuracy evaluation. I don't really see how this can provide any valuable information on accuracy, since you don't know the true origin of assigned fish.

- *This has been expanded upon in the methods. (lines 166-167)*

Line 364-365. There are a few studies that have combined genetics and telemetry, e.g. Östergren et al., 2012.

Figures

Figure 1. what is geographic assignment units? In the ms you use reporting units.

- *The wording has been changed*

Figure 2. Is letters A and B missing to show which is which? i.e. above and below.

- *This has been changed*

Figure 3. Way too small text.

- *This has been changed*

Figure 4. Not sure that reference is need in the figure text.

- *This figure has been removed*

Figure 5. What are the dots??

- *This figure has been removed*

Table 1. A bit confusing table. What is different between fins and fin clip?

- *This column has been removed*

References

Whitlock, R., Mäntyniemi, S., Palm, S., Koljonen, M.-L., Dannewitz, J., Östergren, J. (2018) Integrating genetic analysis of mixed populations with a spatially- explicit population dynamics model. *Methods in Ecology and Evolution*. 2018:1-19 DOI: 10.1111/2041-210X.12946.

Östergren, J., Nilsson, J. & Lundqvist, H. (2012). Linking genetic assignment tests with telemetry enhances understanding of spawning migration and homing in sea trout *Salmo trutta* L. *Hydrobiologia*. (DOI) 10.1007/s10750-012-1063-7. since the main aim of the manuscript is GSI.

- *Where applicable, the method of GSI has been expanded upon, see above. However it was not the intention of the manuscript to focus on GSI, rather this technique is used as a tool to uncover underlying biological patterns (in this case, migration timing) within wild salmon populations.*

Line 118-120. Here you state the aims of the manuscript. There is nothing mentioned about the link to new models for management of sea lice problems in relation to aquaculture. Fine, but you later and in the abstract claim there is??

- *This has now been changed (lines 143-145)*

Appendix C

Review of resubmission

The manuscript "Inferring Atlantic salmon post-smolt migration patterns using genetic assignment" has now been revised and most comments from two reviewers (of which I was one) has been adequately considered. This version of the manuscript is much better. The introduction now has a very nice flow and has been broadened to be more suitable to a more general public. The methods used are nicely introduced and objectives and aim of the study is now clear. Results and discussion are more focused and clear.

I only have a few minor comments. Other than that the ms seems acceptable for publication.

Line 187. I still think that you need to mention *how* the quality check was done. Good to mention that a second person checked the scoring, but how? Did that person re-analyse some data blindly? How much of the data? 10% or 10 individuals? And what were the results? 100% correct or not? Also, I can see in the submitted dataset that there are rather much missing values. Some individuals are missing data in eight microsatellites. There is not mentioned anything about scoring results, missing data and how this is handled. I think this should be calculated and mentioned. For example, was there no threshold for when including an individual?

Line 226. I also think that the visual inspection still is rather weak. But, I am aware of the potential problems of doing statistical tests of this kind of results. A visual inspection, in particular of 95% confidence intervals, could be fine in this case.

Line 344. I agree that the handling and tagging in telemetry studies might affect behavior of tagged fish, but I would like to see a reference here. You might find one in this paper, but here the focus is on that there is no difference in behavior of tagged fish:

<https://onlinelibrary.wiley.com/doi/full/10.1111/j.1095-8649.2006.01101.x>

Figure 1 and 3, and S3-S5 Supporting information. I still think that the legends in the figures are too small. One should not have to zoom in to be able to read...

Appendix D

Review received: 13 August 2019

Re-submitted: 19 August 2019

Response to reviewer comments RSOS-190426.R1

Once again thank you to the editors and reviewers taking the time to go through the paper again and provide further suggestions and feedback. We have carefully answered the editor and reviewer comments, and our responses are provided below.

Associate Editor Comments to Author (Dr Kristina Sefc):

Associate Editor: 1

Comments to the Author:

Both reviewers commend the authors for the good job done with the revision. The reviews identify information that still needs to be added to the manuscript for clarity and offer suggestions for improved wording and structuring. In addition to following these suggestions, please make sure to provide the information on sampling week and the two files (reporting group and fisheries) requested by Reviewer 1. Also, please make sure that the excel table with the microsatellite data contains all necessary information to convert it into the genepop files used in your analyses, or provide the genepop files.

- *All the above has now been addressed, please see details below.*

Reviewer comments to Author:

Reviewer: 1

Comments to the Author(s)

I think the authors did a good job especially with improving the focus of the paper. The manuscript is substantially shorter now, with several parts removed, making it much more concise and readable. I only have a few comments/suggestions left - mainly of editorial nature.

Abstract:

I think the abstract fails to do justice to what for me still is the main resource of the current paper - the genetic baseline for the Hardangerfjord salmon populations. I suggest to add a few facts to the abstract, along the lines of 'established a genetic baseline based on 31 microsatellite markers for the main salmon rivers in the Hardangerfjord. We evaluated the baseline with respect to individual assignment success and MSA...'.. Then it was used for MSA to evaluate the composition of trawl catches, and so on.

Line 51-53 is outdated. In the current version you limit yourself to MSA, which does not involve assigning single individuals back to rivers or regions.

- *The abstract has now been amended as suggested (lines 51-53)*

Introduction:

Line 74: suggested change to: .. closely linked to the fitness and abundance ..

- *This has been changed (line 72)*

Line 84: suggested change to: .. genetic differentiation is low in some species ..

- *This has been changed (line 82)*

Line 116: suggested change to: .. is currently the world's largest producer of farmed salmon, ...

- *This has been changed (line 114)*

Methods:

As it is now there will be two supplementary files. The first is an excel table that contains the details on the microsats, including primers and amplification conditions across a number of tabs (never really referenced in the main manuscript as far as I can see) and the second a word document with supplementary methods (currently referred to as Supplementary file 1). I suggest to move the tables with the Microsat primers and the conditions to Supplementary file 1. This will leave the other supplementary file with just microsatellite data, which I think is a better organization of the supplementary info.

- *The primer & amplification protocols are now referenced in the text (lines 182-183) and have been moved to tables in Supplementary file 1.*

Line 184: change to something like: (for amplification conditions see Supplementary File 1)

- *See above (lines 182-183)*

Line 199: Provide version number of ONCOR software used for the analyses.

- *I have double checked the website and the manual and cannot find a version number for ONCOR.*

Line 214: The stock proportions you used for the realistic fishery simulation were obtained from the literature. Please provide the actual proportions you used as input for the realistic fishery simulation. One can sort of deduce them from fig. S2, but it would be good to provide the actual numbers.

- *This information has been added (line 216) and a fisheries file with the proportions has been included in Supplementary File S2.*

Line: 227. As far as I can see the results from the river level MSA has been removed from the paper. I suggest to remove this sentence or present the result.

- *This has been removed.*

To make their analyses fully reproducible the authors need to also provide the actual stock proportions they used for the realistic fishery simulation. For the trawl microsat data which was used for the MSA I can deduce from the sample name which year it was taken, but not which week. Since the MSA was done week by week this info needs to be provided somewhere (either additional table or extra column) if the analyses should be fully reproducible.

- *The fisheries file and the weekly genepop files are now included in the Supplementary File S2.*

With the extra information discussed above I think the analyses are in principle fully reproducible, so I leave the following to the editor's discretion: The Microsat data is made available as Excel tables, which is fine. I think it would help, though, if the authors would upload also the actual GENEPOP format files (Baseline file, Mixture files - would be for each week) that they used in the ONCOR analyses. I recommend that they also upload the Reporting group file (grouping populations into regional units) and the fisheries file (parametric stock proportions for the realistic fisheries simulations).

- *All the requested files are now included in the Supplementary File S2.*

Results:

Line 244-245: Can't assess the PCA because there are no population labels in the figure S1.

- *I do not know why this is not visible, but I have checked and the population codes are visible in the figure in Supplementary File S1. I attach a screen shot of the figure:*

Figure S1: Outputs of the principle component analysis for (A) the full baseline and (B) the baseline without Oselva (OS) and Tysse (TY).

Line 248: suggested change to: .. found that on average 53.1% of fish were correctly assigned ..

- *This has been changed (line 248)*

Line 252: suggested change to: .. improved to 72.1% on average (Table 2).

- *This has been changed (line 252)*

Line 259: You have removed the analyses regarding 'date of migration from the river' and also the individual based assignment results from the paper, so please update the heading of this section.

- *This has been changed (line 260)*

Line 260: I find the start of this sentence confusing. Guess it must be a remnant of the previous version since all individual based assignment results were removed from the paper now. Since the MSA does not individually assign fish to a region, I suggest to remove the first part of the first sentence, and start it with: The weekly MSA..

- *This has been changed (line 261)*

Discussion:

- I suggest to swap the sections 'Genetic assignment success' and 'Spatio-temporal patterns of post-smolt migration'. That would seem a more logical order.

- *This has been changed (see Discussion)*

Parts of the sections 'Spatio-temporal patterns of post-smolt migration' and 'Conclusions & Practical Implications' are redundant. I suggest to either merge both sections under a new heading or, if the authors want to keep the two sections and the editor agrees, I would essentially move everything starting with the sentence 'Differences in fjord exit times' in line 284 to the 'Practical implications' section and expand/replace lines 330-337 there.

- *This section has now been reworked as suggested (lines 302-338)*

I would also move the last paragraph from the conclusions section to the start of the conclusions section. I think the section is fine, but I find it odd to close the entire manuscript with a call for combining telemetry and genetic methods, but would rather end with the call for incorporating new spatio-temporal data into existing management models, which has been really the theme of the paper under the cover..

- *This has been changed (lines 302-308)*

Figure 1:

Remove the last sentence from the caption. The parts referred to have been removed from the figure.

- *This has been removed.*

Figure 3:

For this figure the last sentence from the caption of figure 1, i.e. the note about the trawls not going over land, would be appropriate.

- *This has been added (Figure caption: Figure 3)*

Supplementary File 1:

Please add labels (river) to the PCA (Fig. S1). I just see blue dots without labels.

- *Please see above, there are blue dots with the population codes in the figure*

Reviewer: 2

Comments to the Author(s)

Review of resubmission

The manuscript "Inferring Atlantic salmon post-smolt migration patterns using genetic assignment" has now been revised and most comments from two reviewers (of which I was one) has been adequately considered. This version of the manuscript is much better. The introduction now has a very nice flow and has been broadened to be more suitable to a more general public. The methods used are nicely introduced and objectives and aim of the study is now clear. Results and discussion are more focused and clear.

I only have a few minor comments. Other than that the ms seems acceptable for publication.

Line 187. I still think that you need to mention *how* the quality check was done. Good to mention that a second person checked the scoring, but how? Did that person re-analyse some data blindly? How much of the data? 10% or 10 individuals? And what was the results? 100% correct or not?

- *Further clarification has been added in the text (line 185-186). For further information: The alleles are scored by one individual, then a second person goes through the files to check that all alleles have been correctly scored (for example, no extra alleles or no missing score where there is an allele). This is done for each individual (here, 1533 baseline individuals genotyped, 814 trawl individuals genotyped (total 2347) for all multiplexes. The scoring procedure is well established and does not usually result in errors, rather if the first individual missed an allele or added an allele by mistake then the second person would remove or add an allele as appropriate. In this case these additions or deletions were not recorded and usually they are negligible (less than 100 for all individuals across all multiplexes).*

Also, I can see in the submitted dataset that there are rather much missing values. Some individuals are missing data in eight microsattelites. There is not mentioned anything about scoring results, missing data and how this is handled. I think this should be calculated and mentioned. For example, was there no threshold for when including an individual?

- *Information on this has now been added (line 187-188)*

Line 226. I also think that the visual inspection still is rather weak. But, I am aware of the potential problems of doing statistical tests of this kind of results. A visual inspection, in particular of 95% confidence intervals, could be fine in this case.

Line 344. I agree that the handling and tagging in telemetry studies might affect behavior of tagged fish, but I would like to see a reference here. You might find one in this paper, but here the focus is on that there is no difference in behavior of tagged fish:

<https://onlinelibrary.wiley.com/doi/full/10.1111/j.1095-8649.2006.01101.x>

- *References have now been included (line 308)*

Figure 1 and 3, and S3-S5 Supporting information. I still think that the legends in the figures are too small. One should not have to zoom in to be able to read...

- *Legend size has now been changed*